# CORAL: Uncertainty-Aware Regulation of Exposure Concentration in Recommender Systems

**Nitin Bisht** [* 1 2]  **Linjiang Guo** [* 1]  **Xiuwen Gong** [3]  **Huan Huo** [1]  **Guandong Xu** [2]

## Abstract

Recommender systems (RS) may suffer from feedback-driven exposure concentration, where repeated engagement optimization collapses exposure onto a narrow set of categories, reducing catalog coverage and degrading long-horizon learning. Existing methods are often post hoc and typically lack principled uncertainty-aware risk estimates for regulating exposure under endogenous feedback. We therefore propose CORAL, a model-agnostic, uncertainty-aware framework that formulates exposure regulation as a constrained sequential decision problem. Specifically, we model self-reinforcing interactions to construct an exposure-saturation state, then derive an upper confidence bound on category-conditioned violation risk from observed history and incorporate it through a state-dependent penalty for adaptive intervention near saturation. Moreover, we provide theoretical guarantees for risk bounds, finite-time recovery, and efficient long-term performance. Extensive experiments on real-world datasets and controlled simulations validate the effectiveness of the proposed framework, which aligns with our theoretical analysis. Our code is available at: https://github.com/downw/CORAL.

## 1. Introduction

Recommendations shape the recommender system's (RS) observed feedback and, in turn, its future learning signal (Jiang et al., 2019). While much effort goes into improving user utility(Nguyen et al., 2014; Chen et al., 2025), comparatively less attention has been paid to the long-run reliability of RS under endogenous feedback. Repeated optimization

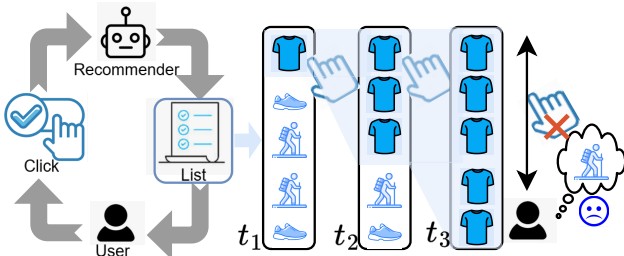

*Figure 1.* **Recommendation collapse in the feedback loop.** Left: The interaction between the recommender system and the user. Right: Over time ($t_1 \rightarrow t_3$), repeated clicks on a single category (e.g., T-shirts) lead the list to collapse into a single category, filtering out diverse interests like hiking gear and degrading user experience in the long-term.

for short-term utility can amplify this effect, progressively concentrating exposure on a narrow subset of categories, a phenomenon often leading to echo-chamber effects in practice (Ziegler et al., 2005; Zhao et al., 2025). This is not only a diversity concern but also a structural learning failure. As exposure collapses and catalog coverage vanishes, the data required for long-horizon user modeling becomes unavailable, leading to eventual model degeneracy (Jiang et al., 2019; Deffayet et al., 2023). As illustrated in Figure 1, once exposure concentration compounds, coverage collapses, and long-horizon learning degrade, making reliable adaptation increasingly difficult. In recent years, some methods such as post hoc diversification and re-ranking constraints (Balloccu et al., 2022), long-horizon exploration objectives (Pitis et al., 2020; Gao et al., 2022), and causal correction for exposure bias (Gao et al., 2024; Li et al., 2025) have been proposed. However, these approaches are typically motivated by diversity or unbiased estimation goals, rather than closed-loop control of long-run exposure dynamics, and thus do not directly prevent feedback-induced exposure collapse under endogenous feedback. Moreover, they typically lack principled uncertainty quantification for the risk of violating a critical exposure threshold under stochastic feedback, making it unclear when and how strongly to intervene.

As a result, we are motivated to develop a recommender framework with statistical guarantees for risk-aware regulation of exposure dynamics under endogenous interaction.

---

[*]Equal contribution  [1]University of Technology, Sydney [2]The Education University of Hong Kong [3]School of Computer Science, Wuhan University, Wuhan, China. Correspondence to: Xiuwen Gong <gongxiuwen@gmail.com>, Huan Huo <Huan.Huo@uts.edu.au>, Guandong Xu <gdxu@eduhk.hk>.

*Proceedings of the 43rd International Conference on Machine Learning*, Seoul, South Korea. PMLR 306, 2026. Copyright 2026 by the author(s).

Our overall goal is to prevent feedback-induced exposure concentration while maintaining high recommendation utility. Thus, the objectives of our framework is threefold: (1) to effectively balance between recommendation utility and exposure diversity; (2) to control exposure-saturation violations to a platform-defined tolerance level under stochastic user responses; and (3) to intervene adaptively, i.e., applying minimal regulation when far from the threshold and increasing regulation only as saturation risk grows, to avoid unnecessary utility loss.

Inspired by safe online learning and constrained decision-making under uncertainty (Gu et al., 2024), we propose **CORAL** (**C**onstrained **O**ptimistic **R**isk-**A**ware **L**earning), a risk-aware control framework for sequential recommendation that regulates exposure dynamics. However, safety is defined not by an immediate observable cost after each recommendation, but by whether a history-dependent exposure process crosses a critical threshold under endogenous feedback. As a result, these methods, in their natural form do not directly address our key objectives. Specifically: (1) how to define and estimate a user-specific exposure state from interaction history that captures self-reinforcing concentration; (2) how to obtain statistically grounded uncertainty bounds on exposure-saturation risk under stochastic user responses; and (3) how to convert the estimated exposure state and risk bounds into a principled, state-dependent regulation schedule that is also adaptive. To address these challenges, we first model user interactions with a Hawkes-inspired intensity model (Wen & Fang, 2022) to capture self-reinforcing exposure dynamics, and construct an exposure saturation state from the estimated intensities. We then formalize saturation as a threshold-violation event and derive an uncertainty-aware upper confidence bound on the category-conditioned violation risk under stochastic user responses, enabling reliable regulation without relying on single-point estimates. Finally, we integrate this risk bound into the decision rule via a state-dependent penalty that increases as the saturation state approaches the threshold, resulting in gradual, proportional intervention while preserving recommendation utility. The overall framework is depicted in Figure 2. We summarize our contributions as follows:

- We firstly formulate feedback-driven exposure concentration in recommender systems as a constrained control-and-learning problem, maximizing cumulative utility while controlling saturation risk that can collapse coverage and learnability.
- Secondly, we propose a model-agnostic framework, **CORAL**, that regulates exposure dynamics under endogenous feedback. CORAL models self-reinforcing interactions via a Hawkes-inspired intensity model to construct an exposure saturation state, and derives a confidence-aware upper bound on category-

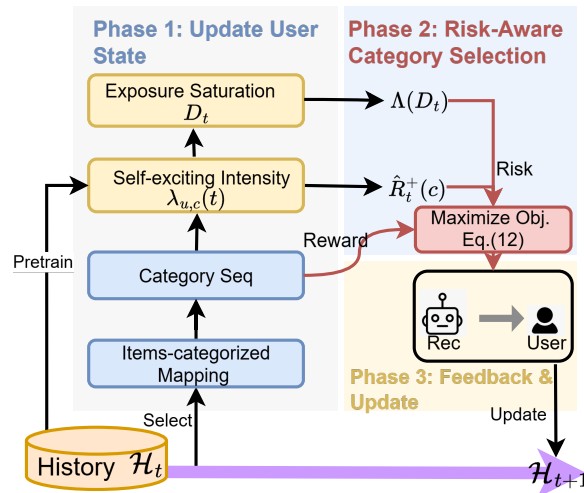

*Figure 2.* Overview of the proposed framework. We (i) update the user state (intensity and saturation), (ii) perform risk-aware category selection, and (iii) process feedback and update the history.

conditioned saturation risk from observed violations, and incorporates this bound as a decision rule via a state-dependent penalty on top of standard ranking backbones.

- Next, we establish rigorous theoretical results for CORAL. We show that the regulation mechanism recovers in finite time, ensuring stability (Theorem 2), and that CORAL achieves long-run performance with sublinear utility regret over the horizon (Theorem 3), which theoretically verifies effectiveness of CORAL.

- Finally, we conduct comprehensive experiments across both real-world datasets and controlled simulations, demonstrating the empirical effectiveness and efficiency of the proposed CORAL, which aligns with the theoretical analysis.

## 2. Problem Setup

We first introduce the notations used in this paper. Let $\mathcal{U}$ and $\mathcal{I}$ denote the set of users and items, respectively. To capture diverse user interests, each item $a \in \mathcal{I}$ is associated with a specific content category $c(a) \in \mathcal{C}$. For each category $c \in \mathcal{C}$, let $\mathcal{I}_c := \{a \in \mathcal{I} : c(a) = c\}$ denote the set of items belonging to category $c$. We consider a sequential recommendation setting where, at each time step $t \in \{1, \dots, T\}$, the policy $\pi$ selects an action $a_t \in \mathcal{I}$ and observes feedback $y_t \in \mathcal{Y}$, receiving an immediate reward $r(a_t, y_t) \in [0, 1]$. We denote the interaction history up to time $t$ by $\mathcal{H}_t := \{(a_1, y_1), \dots, (a_t, y_t)\}$. Conventionally, the objective is to learn a policy $\pi^*$ that maximizes cumulative expected utility over horizon $T$. However, unconstrained maximization amplifies echo chamber effects, leading to RS degeneracy as the model progressively narrows

exposure distribution based on its own past recommendations. To prevent this irreversible collapse of learnability, we reformulate the task as a constrained optimization problem. We define an exposure-load state $D_t$, computed from the learned intensity model, representing the user's cumulative exposure load. The ideal objective is to regulate the long-run state of the exposure process such that this saturation never exceeds a critical degeneracy threshold $\lambda_{\max}$:

$$\max_{\pi} \mathbb{E}\left[\sum_{t=1}^{T} r(a_t, y_t)\right]$$
$$\text{s.t. } \Pr(D_{t+1} > \lambda_{\max} \mid \mathcal{H}_{t-1}) \le \delta, \quad \forall t \in \{1, \ldots, T\}. \tag{1}$$

where $\delta \in (0, 1)$ is a pre-defined tolerance for risk.

## 3. The Proposed Framework

While Equation 1 provides a theoretical objective, directly enforcing it is intractable in a real-world streaming setting for two critical reasons. Firstly, $D_t$ is regulated by latent user dynamics and a static model cannot capture the complex nature of exposure concentration. Secondly, the feedback loop creates temporal correlations where the future risk depends on the history $\mathcal{H}_t$. Since the probability $\Pr(D_{t+1} > \lambda_{\max} \mid \mathcal{H}_t)$ is unknown and non-stationary, the constraint cannot be evaluated in closed form. To resolve this, we propose our novel framework CORAL.

We begin by capturing the user's self-reinforcing exposure concentration. To achieve this, we introduce a Hawkes-inspired self-exciting intensity model for the user's interaction intensity $\lambda_{u,c}(t)$ towards a specific category $c$. We express the interaction intensity as a user history-dependent function where the user's past consumption events probabilistically excite the future actions. Specifically:

$$\lambda_{u,c}(t) = \mu_{u,c} + \sum_{k=1}^{t-1} \alpha_{c,c(a_k)} \cdot \exp(-\beta_{u,c}(t-k)) \, e_k, \tag{2}$$

where $c(a_k)$ denotes the content category of item $a_k$ selected at time step $k$, and $e_k \in \{0, 1\}$ is an engagement *event indicator* derived from feedback $y_k$ using:

$$e_k := \mathbb{I}(g(y_k) \ge \tau), \tag{3}$$

for a defined mapping $g(\cdot)$ and threshold $\tau$. Furthermore, $\mu_{u,c} \ge 0$ represents the user's base rate, capturing their homeostatic preference for category $c$ in the absence of external stimuli; $\beta_{u,c} > 0$ denotes the resilience, quantifying the temporal decay rate at which the urge to consume $c$ dissipates; and $\alpha_{c,c(a_k)}$ is the pairwise susceptibility coefficient. This coefficient corresponds to the entry $(c, c(a_k))$ in the susceptibility matrix $\boldsymbol{A}_u = [\alpha_{c,c'}]$, quantifying the excitation intensity triggered in category

$c$ by a past event of category $c'$. It aggregates the influence of the user's entire interaction history. To ensure the framework adapts to individual users, we estimate these stability parameters $\theta_u = (\mu_u, \boldsymbol{A}_u, \beta_u)$ via maximum likelihood estimation (MLE) on the user's history $\mathcal{H}_u := \{(a_1, y_1), \ldots, (a_T, y_T)\}$. We minimize the negative log-likelihood over the horizon $t = 1, \ldots, T$:

$$\mathcal{L}(\theta_u) = -\sum_{t=1}^{T} e_t \log \lambda_{u,c(a_t)}(t) + \sum_{t=1}^{T} \sum_{c \in \mathcal{C}} \lambda_{u,c}(t). \tag{4}$$

Equation (4) follows the multivariate point-process log-likelihood under unit-time discretization. By minimizing it, we estimate $\theta_u = (\mu_u, \boldsymbol{A}_u, \beta_u)$ offline and update $\lambda_{u,c}(t)$ online via Eq. (2), establishing a grounded baseline of the user's stability profile.

Based on the learned intensity model, we can now precisely define the state $D_t$ introduced in Eq. 1. We presume that high intensity is not inherently harmful; it becomes problematic only when it represents a "toxic burst" that deviates significantly from the user's normal baseline.

**Definition 1.** *The Exposure Saturation $D_t$ is defined as the cumulative weighted excess intensity:*

$$D_t = \sum_{c \in \mathcal{C}} w_c \cdot \max\left(0, \lambda_{u,c}(t) - \gamma \cdot \mu_{u,c}\right), \tag{5}$$

*where $w_c \in [0, 1]$ denotes the normative severity weight provided by the platform policy, and $\gamma \ge 1$ is a tolerance factor.*

This state $D_t$ is directly computable from the learned intensities and is used to enforce the safety constraint in Eq. (1).

Having defined the observable state $D_t$, we now define the risk $R_t(c)$ as the conditional probability that selecting category $c$ will push the system into a saturated state:

$$R_t(c) = \Pr(D_{t+1} > \lambda_{\max} \mid \mathcal{H}_{t-1}, \, c(a_t) = c), \tag{6}$$

where $\lambda_{\max}$ is a pre-defined safety threshold. Since the user's response $y_t$, which triggers the excitation, is stochastic, the future state $D_{t+1}$ is random. Since $R_t(c)$ cannot be observed directly, we need to estimate it from the history of empirical violations. Specifically, we define the instantaneous loss $L_{t+1}$ as the indicator of a safety violation resulting from the realized action at step $t$:

$$L_{t+1} = \mathbb{I}(D_{t+1} > \lambda_{\max}). \tag{7}$$

To quantify the uncertainty of our estimate, we examine the prediction residual as follows:

$$M_{t+1} = L_{t+1} - R_t(c(a_t)). \tag{8}$$

The following lemma validates that this sequence of residuals forms a Martingale Difference Sequence (MDS).

**Lemma 1.** *Let $\mathcal{G}_t := \sigma(\mathcal{H}_{t-1}, a_t)$ be the sigma-algebra generated by the interaction history up to time $t-1$ and the realized action at time $t$. Given $R_t(c(a_t)) = \mathbb{E}[L_{t+1} \mid \mathcal{G}_t]$, then the residual $\{M_{t+1}\}_{t \geq 1}$ constitutes a Martingale Difference Sequence (MDS) with respect to $\{\mathcal{G}_t\}$, satisfying:*

$$\mathbb{E}[M_{t+1} \mid \mathcal{G}_t] = 0.$$

*Proof.* See Section E.1. □

This result implies that the discrepancy between the observed $L_{t+1}$ and the conditional risk is driven by randomness in the user response. Consequently, we can bound the latent risk by explicitly accounting for uncertainty:

---

**Algorithm 1** The CORAL Algorithm
---
1: **Input:** User $u$, catalog $\mathcal{I}$, categories $\mathcal{C}$, Safety Limit $\lambda_{\max}$, Risk Budget $\delta$, Exploration $\kappa$, constants $\epsilon > 0$, $\Lambda_{\max} > 0$.
2: **Initialize:** For all $c \in \mathcal{C}$: $N_0(c) = 0$, $\hat{r}_0(c) = 0$; History $\mathcal{H}_0 = \emptyset$.
3: **for** $t = 1, 2, \ldots, T$ **do**
4:     // Step 1: Update User State
5:     Update intensity $\lambda_{u,c}(t)$ and exposure saturation $D_t$ from history $\mathcal{H}_{t-1}$.     [Def. 1]
6:     Compute adaptive penalty $\Lambda(D_t)$.     [Eq. 11]
7:     // Step 2: Risk-Aware Category Selection
8:     Compute risk bounds $\hat{R}_t^+(c)$ for all $c \in \mathcal{C}$.
9:     Select category $c_t$.     [Eq.12].
10:     Select item $a_t \in \arg\max_{a \in \mathcal{I}_{c_t}} s_t(a)$, where $s_t(a)$ is a base recommender score.
11:     // Step 3: Feedback & Update
12:     Recommend $a_t$, observe feedback $y_t$, set $\rho_t = r(a_t, y_t)$, and compute violation $L_{t+1} = \mathbb{I}(D_{t+1} > \lambda_{\max})$.
13:     Update statistics:
14:        $N_{t+1}(c_t) \leftarrow N_t(c_t) + 1$
15:        $\hat{r}_{t+1}(c_t) \leftarrow \hat{r}_t(c_t) + \frac{1}{N_{t+1}(c_t)}(\rho_t - \hat{r}_t(c_t))$
16:     Update history $\mathcal{H}_t \leftarrow \mathcal{H}_{t-1} \cup \{(a_t, y_t)\}$.
17: **end for**

---

**Theorem 1.** *For any category $c \in \mathcal{C}$, let $N_t(c) := \sum_{k=1}^{t-1} \mathbb{I}(c(a_k) = c)$ and the empirical violation rate $\hat{R}_t(c) := \frac{1}{N_t(c)} \sum_{k=1}^{t-1} \mathbb{I}(c(a_k) = c) L_{k+1}$. Let $\mathcal{G}_k := \sigma(\mathcal{H}_{k-1}, a_k)$ and we define the historical average conditional violation risk as $\bar{R}_t(c) := \frac{1}{N_t(c)} \sum_{k=1}^{t-1} \mathbb{I}(c(a_k) = c) \mathbb{E}[L_{k+1} \mid \mathcal{G}_k]$. Then for any $\delta \in (0, 1)$, with probability at least $1 - \delta$, the following holds simultaneously for all*

$t \leq T$ *and all* $c \in \mathcal{C}$ *with* $N_t(c) > 0$:

$$\bar{R}_t(c) \leq \hat{R}_t(c) + \sqrt{\frac{2\,\hat{R}_t(c)\big(1 - \hat{R}_t(c)\big)\,\ln\left(\frac{2T|\mathcal{C}|}{\delta}\right)}{N_t(c)}}$$
$$+ \frac{3\,\ln\left(\frac{2T|\mathcal{C}|}{\delta}\right)}{N_t(c)} =: \hat{R}_t^+(c). \qquad (9)$$

*Proof.* See Section E.2. □

**Remark 1.** *In Theorem 1, we derive an adaptive upper bound $\hat{R}_t^+(c)$ on the history-averaged category-conditioned violation risk $\bar{R}_t(c)$. It provides a conservative estimate of violation probability from observed history, which CORAL combines with the current-state multiplier $\Lambda(D_t)$ in Eq. (12) for risk-aware decision-making under uncertainty.*

### 3.1. Decision Policy

We utilize the safety bound established in Theorem 1 to construct a risk-aware decision policy. Specifically, we use $\hat{R}_t^+(c)$ to balance utility and risk through a state-dependent penalty to define the category-level objective:

$$\mathcal{J}_t(c) = \mathbb{E}[r(c, y_t) \mid \mathcal{H}_{t-1}] - \Lambda(D_t) \cdot \hat{R}_t^+(c), \qquad (10)$$

where $\mathbb{E}[r(c, y_t) \mid \mathcal{H}_{t-1}]$ denotes the expected reward of selecting an item from category $c$ at time $t$ and $\Lambda(D_t)$ is the adaptive penalty function given by:

$$\Lambda(D_t) = \min\left\{\Lambda_{\max}, \frac{1}{\max(\epsilon, \lambda_{\max} - D_t)}\right\}, \qquad (11)$$

with $\epsilon > 0$ a constant ensuring stability and $\Lambda_{\max} > 0$ a cap to prevent numerical instability near the threshold. When $D_t \ll \lambda_{\max}$, $\Lambda(D_t)$ is small, it allows the policy to focus on maximizing user satisfaction. Conversely, as the exposure approaches the critical limit, i.e., $D_t \to \lambda_{\max}$, $\Lambda(D_t)$ increases and saturates at $\Lambda_{\max}$.

**Decision Policy Optimization.** Since the expected reward in Eq. (10) is unknown, we apply the Optimism in the Face of Uncertainty (OFU) principle and optimize:

$$c_t = \arg\max_{c \in \mathcal{C}} \left[\hat{r}_t(c) + \kappa\sqrt{\frac{\ln t}{\max(1, N_t(c))}}\right.$$
$$\left. - \Lambda(D_t)\hat{R}_t^+(c)\right]. \qquad (12)$$

Here, $\hat{r}_t(c) := \frac{1}{\max(1, N_t(c))} \sum_{k=1}^{t-1} \mathbb{I}(c(a_k) = c) \rho_k$ denotes the empirical mean reward, where $\rho_k := r(a_k, y_k)$ is the realized reward at time $k$, and $N_t(c) := \sum_{k=1}^{t-1} \mathbb{I}(c(a_k) = c)$ is the number of times category $c$ was selected up to time $t - 1$. The parameter $\kappa > 0$ controls exploration. After

selecting $c_t$, we choose the item $a_t \in \mathcal{I}_{c_t}$ using the underlying recommender's ranking score $s_t(a)$ within the chosen category. Maximizing (12) balances optimistic utility and the risk penalty and therefore provides a tractable solution to Equation (1).

**Inference.** At each step $t$, CORAL updates the intensity $\lambda_{u,c}(t)$ and exposure saturation $D_t$ from the pre-decision history $\mathcal{H}_{t-1}$, quantifying the user's current saturation state. Next, using Eq. 12, it resolves the tradeoff between the optimistic utility estimate $\hat{r}_t(c)$ and the risk penalty $\Lambda(D_t)\hat{R}_t^+(c)$. When exposure saturation is low ($D_t \ll \lambda_{\max}$), $\Lambda(D_t)$ is small and utility dominates. As $D_t$ approaches the critical limit $\lambda_{\max}$, $\Lambda(D_t)$ increases and saturates at $\Lambda_{\max}$, amplifying the penalty on risky categories and shifting selection toward low-risk alternatives. After selecting $c_t$, CORAL selects $a_t := \arg\max_{a\in\mathcal{I}_{c_t}} s_t(a)$ and recommends it; it then observes feedback $y_t$, sets $\rho_t := r(a_t, y_t)$, and updates the history to $\mathcal{H}_t := \mathcal{H}_{t-1} \cup \{(a_t, y_t)\}$.

# 4. Theoretical Analysis

In this section, we provide the theoretical analysis of our CORAL framework. We validate it through *stability*, ensuring finite-time restoration of safety constraints, and *efficiency*, ensuring it converges to the optimal policy. Our analysis relies on standard assumptions that (1) rewards $r \in [0, 1]$ and Hawkes parameters are bounded; and (2) the risk estimation noise constitutes a sub-Gaussian Martingale difference sequence.

## 4.1. Stability

A critical challenge in safe sequential recommendation is avoiding permanent saturation, where a user's exposure load $D_t$ remains indefinitely above the safety limit $\lambda_{max}$. We first prove that CORAL guarantees a strictly finite duration for these regulation phases.

**Theorem 2.** *Let $t_0$ be a time step such that $D_{t_0} \geq \lambda_{\max} - \epsilon$ for some $\epsilon > 0$. Assume that during regulation, the selected category induces residual aggregate excitation bounded by $\alpha_{\mathrm{safe}}$ after Hawkes decay, so that $D_{t+1} \leq e^{-\beta_{\min}} D_t + \alpha_{\mathrm{safe}}$, where $\beta_{\min} := \min_{c\in\mathcal{C}} \beta_{u,c}$. If this residual excitation is below the decay margin at the safety boundary, i.e., $\alpha_{\mathrm{safe}} < (1 - e^{-\beta_{\min}})(\lambda_{\max} - \epsilon)$, and the penalty cap $\Lambda_{\max}$ is chosen large enough that, whenever $D_t \geq \lambda_{\max} - \epsilon$, the optimizer of Eq. (12) selects such a low-excitation category, then the exposure load decays to a safe state within finite time. In particular, the regulation phase length $\tau_{\mathrm{reg}}$ satisfies:*

$$\tau_{\mathrm{reg}} \leq \left\lceil \frac{1}{\beta_{\min}} \ln\left( \frac{D_{t_0} - \frac{\alpha_{\mathrm{safe}}}{1-e^{-\beta_{\min}}}}{(\lambda_{\max} - \epsilon) - \frac{\alpha_{\mathrm{safe}}}{1-e^{-\beta_{\min}}}} \right) \right\rceil. \quad (13)$$

*Proof.* See Section E.3. □

**Remark 2.** *Theorem 2 theoretically validates the "Anti-Echo Chamber" mechanism under a low-excitation regulation condition. By proving that $\tau_{\mathrm{reg}}$ is strictly finite, we confirm that the framework effectively interrupts feedback loops and prevents the system from remaining permanently trapped in a saturated state.*

**Corollary 1.** *Under the neutral-category special case $\alpha_{\mathrm{safe}} = 0$, i.e., when the selected regulation category has zero aggregate excitation, Theorem 2 reduces to:*

$$\tau_{\mathrm{reg}} \leq \left\lceil \frac{1}{\beta_{\min}} \ln\left( \frac{D_{t_0}}{\lambda_{\max} - \epsilon} \right) \right\rceil.$$

*Proof.* See Section E.4. □

## 4.2. Regret and Learning Efficiency

Having guaranteed stability, we now quantify the learning efficiency. We define the *Sustainable Regret* $\mathfrak{R}(T)$ as the utility gap between CORAL and an oracle-safe comparator induced by the optimal safe Oracle $\pi^*$.

**Theorem 3.** *Let $\Delta_{\max} \in [0, 1]$ be the maximum instantaneous reward gap and $N$ denote the number of regulation episodes triggered up to time $T$, where the $k$-th episode has duration $\tau_{\mathrm{reg}}^{(k)}$. Let $\mathcal{T}_{\mathrm{nor}}$ denote the non-regulation rounds, and let $c^\star$ be an oracle-safe comparator category. Given its cumulative proxy-risk penalty as $B_T(c^\star) := \sum_{t\in\mathcal{T}_{\mathrm{nor}}} \Lambda(D_t)\hat{R}_t^+(c^\star)$. Then, with probability at least $1 - \delta$, the cumulative regret satisfies:*

$$\mathfrak{R}(T) \leq O\left( \kappa\sqrt{|\mathcal{C}|T\ln T} \right) + B_T(c^\star) + \Delta_{\max}\sum_{k=1}^{N} \tau_{\mathrm{reg}}^{(k)}.$$

*In particular, when the oracle-safe comparator has sublinear cumulative proxy-risk, i.e., $B_T(c^\star) = \widetilde{O}(\sqrt{T})$, the regret remains sublinear up to the finite regulation cost.*

*Proof.* See Section E.5. □

**Remark 3.** *Theorem 3 validates that CORAL remains an efficient learner under the penalized category-selection rule. The regret consists of a category-level exploration term, the proxy-risk penalty of an oracle-safe comparator, and the transient cost incurred during finite regulation intervals.*

# 5. Experiments

In this section, we conduct experiments to evaluate the effectiveness of the proposed CORAL framework. Specifically, we design experiments to **(1)** evaluate CORAL on real-world sequential recommendation benchmarks under step-by-step offline evaluation, comparing against representative

*Table 1.* Exposure safety (EC and PS) comparison on Amazon, ML-1M and Steam datasets using the backbones. Our results are in **bold**. Results are averaged over 5 seeds; $p < 0.05$ by a paired two-sided test vs the best competing method.

| Backbone | Method | Metric | Amazon | | | ML-1M | | | Steam | | |
|---|---|---|---|---|---|---|---|---|---|---|---|
| | | | Overall ↓ | Top 20% ↓ | Top 10% ↓ | Overall ↓ | Top 20% ↓ | Top 10% ↓ | Overall ↓ | Top 20% ↓ | Top 10% ↓ |
| Bert4Rec | Naive | EC | 0.46 | 0.87 | 0.89 | 0.24 | 0.41 | 0.52 | 0.29 | 0.38 | 0.44 |
| | | PS | 1.25 | 1.34 | 1.37 | 1.31 | 1.38 | 1.42 | 1.28 | 1.16 | 1.22 |
| | Filter | EC | 0.44 | 0.83 | 0.86 | 0.22 | 0.38 | 0.49 | 0.27 | 0.35 | 0.39 |
| | | PS | 1.12 | 1.21 | 1.25 | 1.18 | 1.24 | 1.29 | 1.15 | 1.05 | 1.10 |
| | TD-VAE-CF | EC | 0.42 | 0.85 | 0.88 | 0.21 | 0.34 | 0.43 | 0.25 | 0.32 | 0.36 |
| | | PS | 0.78 | 0.94 | 1.02 | 0.72 | 0.96 | 1.04 | 0.84 | 0.76 | 0.81 |
| | NuGE | EC | 0.29 | 0.58 | 0.60 | 0.22 | 0.39 | 0.47 | 0.26 | 0.35 | 0.39 |
| | | PS | 0.84 | 1.05 | 1.12 | 0.81 | 1.09 | 1.14 | 0.92 | 0.88 | 0.95 |
| | RL | EC | 0.39 | 0.86 | 0.91 | 0.23 | 0.40 | 0.51 | 0.32 | 0.44 | 0.49 |
| | | PS | 1.19 | 1.26 | 1.29 | 1.24 | 1.27 | 1.31 | 1.21 | 1.11 | 1.18 |
| | **CORAL** | EC | **0.18** | **0.25** | **0.24** | **0.17** | **0.23** | **0.21** | **0.22** | **0.23** | **0.26** |
| | | PS | **0.81** | **0.76** | **0.73** | **0.83** | **0.79** | **0.76** | **0.89** | **0.82** | **0.79** |
| SASRec | Naive | EC | 0.58 | 0.96 | 0.96 | 0.37 | 0.65 | 0.73 | 0.32 | 0.45 | 0.51 |
| | | PS | 1.31 | 1.38 | 1.39 | 1.36 | 1.40 | 1.41 | 1.34 | 1.22 | 1.30 |
| | Filter | EC | 0.57 | 0.93 | 0.95 | 0.30 | 0.55 | 0.61 | 0.28 | 0.37 | 0.41 |
| | | PS | 1.18 | 1.25 | 1.28 | 1.14 | 1.21 | 1.28 | 1.19 | 1.09 | 1.17 |
| | TD-VAE-CF | EC | 0.55 | 0.84 | 0.87 | 0.23 | 0.44 | 0.52 | 0.26 | 0.32 | 0.35 |
| | | PS | 0.81 | 1.05 | 1.12 | 0.75 | 1.07 | 1.13 | 0.86 | 0.80 | 0.85 |
| | NuGE | EC | 0.31 | 0.61 | 0.62 | 0.27 | 0.49 | 0.57 | 0.29 | 0.38 | 0.40 |
| | | PS | 0.92 | 1.12 | 1.18 | 0.85 | 1.16 | 1.22 | 0.95 | 0.98 | 1.06 |
| | RL | EC | 0.44 | 0.87 | 0.88 | 0.34 | 0.60 | 0.69 | 0.35 | 0.49 | 0.54 |
| | | PS | 1.27 | 1.31 | 1.35 | 1.27 | 1.30 | 1.34 | 1.28 | 1.19 | 1.26 |
| | **CORAL** | EC | **0.20** | **0.29** | **0.28** | **0.26** | **0.33** | **0.30** | **0.24** | **0.25** | **0.25** |
| | | PS | **0.94** | **0.91** | **0.89** | **0.87** | **0.83** | **0.82** | **0.96** | **0.92** | **0.90** |

intervention baselines in terms of recommendation utility and exposure saturation metrics, **(2)** assess long-horizon behavior in a closed-loop interactive environment, **(3)** study the robustness of the method from both the modeling and environment perspectives, **(4)** evaluate robustness to category structure under coarse/fine taxonomies and held-out categories, **(5)** analyze sensitivity to the control and risk parameters (Section D.1 in the *Appendix*), and **(6)** conduct an ablation study to disentangle the contributions of core components (Section D.2 in the *Appendix*). We elaborate on the implementation details in Section C.

### 5.1. Datasets and Baseline Methods

We evaluate CORAL across both three diverse real-world recommendation benchmarks and controlled synthetic environments. For real-world evaluation, we use three publicly available datasets: Amazon (He & McAuley, 2016), Steam (Kang & McAuley, 2018), and ML-1M (ML-1M) (Harper & Konstan, 2015). Table 7 reports dataset statistics (users, items, interactions, average sequence length, and density). For comparison, we consider diverse representative intervention methods. Specifically, we compare against: (i) Naive (no mitigation)[1], (ii) Filter

---

[1]acts as an upper threshold for short horizon utility.

(Antikacioglu & Ravi, 2017) (Similarity-based hard safety filtering), (iii) TD-VAE-CF (Gao et al., 2022) (long-term latent interest modeling), (iv) NuGE (Wang et al., 2024) (targeted diversification), and (v) RL (Li et al., 2023) (controllable recommender), alongside our method CORAL. We instantiate all the methods on the two backbones, SAS-Rec (Kang & McAuley, 2018) and Bert4Rec (Sun et al., 2019), and use the same backbone across experiments for fair comparison. Full implementation details, including the closed-loop interaction protocol, sampling/splitting, and baselines, are provided in Appendix C.

### 5.2. Experiment Results

5.2.1. PERFORMANCE ON REAL-WORLD DATASETS

We evaluate exposure safety (EC, PS) and recommendation utility (Recall@10, MRR@10) on real-world recommendation datasets under standard offline evaluation. Here, EC and PS capture how strongly recommendations concentrate and saturate a user's exposure (lower is better), while Recall@10 and MRR@10 quantify retrieval quality (higher is better). Table 1 reports exposure safety (overall and for the Top 20% / Top 10% most over-saturated users identified under Naive), and Table 2 reports utility on the same SASRec and Bert4Rec baselines. These results lead to the following key observations:

*Table 2.* Utility comparison on three datasets. The best results are in **bold** and the second best are underlined. Time is reported in seconds. Results are averaged over 5 seeds; $^*$ indicates $p < 0.01$ by a paired two-sided test vs the second best.

| Backbone | Method | Amazon | | | ML-1M | | | Steam | | |
|---|---|---|---|---|---|---|---|---|---|---|
| | | R@10↑ | M@10↑ | Time(s) | R@10↑ | M@10↑ | Time(s) | R@10↑ | M@10↑ | Time(s) |
| SASRec | Naive | 0.1731 | 0.0733 | 45 | 0.1916 | 0.0660 | 12 | 0.1950 | 0.1211 | 28 |
| | Filter | 0.0837 | 0.0377 | 1844 | 0.1528 | 0.0517 | 85 | 0.1017 | 0.0401 | 332 |
| | TD-VAE-CF | 0.1489 | 0.0634 | 2538 | 0.1347 | 0.0459 | 117 | 0.1306 | 0.1077 | 457 |
| | NuGE | 0.0367 | 0.0084 | 804 | 0.1586 | 0.0446 | 37 | 0.1260 | 0.0408 | 145 |
| | RL | 0.0610 | 0.0248 | 1297 | 0.1297 | 0.0440 | 69 | 0.1259 | 0.0496 | 269 |
| | CORAL | **0.1667**$^*$ | **0.0683**$^*$ | 579 | **0.1768**$^*$ | **0.0601**$^*$ | 57 | **0.1821**$^*$ | **0.1190**$^*$ | 163 |
| Bert4Rec | Naive | 0.1452 | 0.0615 | 25 | 0.1689 | 0.0516 | 11 | 0.1755 | 0.1080 | 14 |
| | Filter | 0.0650 | 0.0284 | 1529 | 0.1310 | 0.0410 | 83 | 0.0890 | 0.0352 | 321 |
| | TD-VAE-CF | 0.1213 | 0.0528 | 2119 | 0.1192 | 0.0395 | 106 | 0.1150 | 0.0917 | 447 |
| | NuGE | 0.0281 | 0.0065 | 582 | 0.1355 | 0.0382 | 39 | 0.1102 | 0.0368 | 137 |
| | RL | 0.0514 | 0.0210 | 1176 | 0.1105 | 0.0369 | 64 | 0.1128 | 0.0423 | 267 |
| | CORAL | **0.1386**$^*$ | **0.0588**$^*$ | 519 | **0.1580**$^*$ | **0.0493**$^*$ | 49 | **0.1610**$^*$ | **0.1025**$^*$ | 173 |

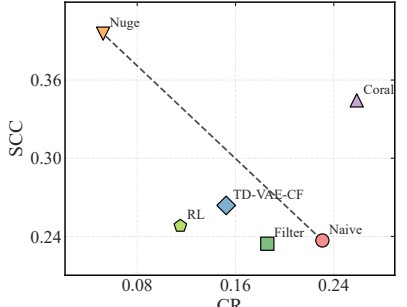 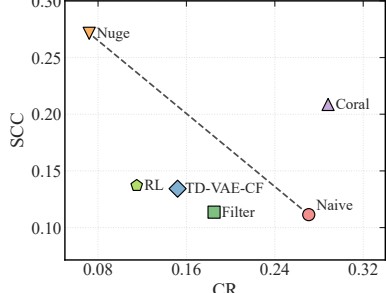 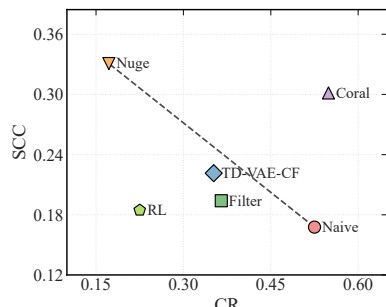

*Figure 3.* Cumulative Reward (CR) vs Sequence Category Coverage (SCC) on Amazon, ML-1M, Steam (left to right) across 100 steps.

- CORAL consistently improves the safety–utility trade-off across datasets and backbones as depicted in Tables 1 and 2. While CORAL incurs a slight drop in short-term utility metrics (e.g., R@10 in Table 2), this is an expected and necessary short-term cost for breaking the feedback loop. It attains on average the lowest EC and PS on Amazon, Steam, and ML-1M under both SASRec and Bert4Rec, and the gains persist for the high-risk tail (Top 20% / Top 10%). Importantly, it also demonstrates strong computational efficiency compared with the other methods in Table 2.

- The unconstrained Naive is a reference for short-horizon utility (Table 2) as it directly optimizes relevance under the offline objective without any exposure regulation; however, the same unconstrained exploitation is exactly what leads to the poor EC/PS behavior (and especially the heavy tail) in Table 1.

- Methods such as Filter reduce concentration primarily by removing items that are "too similar" to a user's history. However such global approach doesn't distinguishes between benign repetition and genuine saturation and often discards high-relevance candidates,

leading to large drops in Recall/MRR as observed in Table 2.

- Structured diversification and nudging methods such as TD-VAE-CF and NuGE aim to diversify exposure through representation-level perturbations or targeted nudges. However, Table 1 shows that their improvements are less consistent among the most oversaturated users, suggesting that fixed diversification mechanisms can under-intervene or over-intervene.

- RL method also struggles to match CORAL on safety and tail risk. While RL optimizes a long-horizon objective, Table 1 indicates it does not reliably reduce EC/PS, especially for the most concentrated users, suggesting optimizing a generic control objective is insufficient without an state-dependent saturation signal.

- Overall, the two tables jointly support the claim that effective mitigation requires a selective, state-dependent regulation mechanism like CORAL.

### 5.2.2. LONG-HORIZON DYNAMICS: UTILITY VS. CONFINEMENT

We evaluate CORAL in closed-loop online environment to test whether it can break echo-chamber confinement without sacrificing long-horizon utility. Following our protocol, we report two complementary metrics: CR (Cumulative Reward, which is used to evaluate total long-horizon utility) and Sequence Category Coverage (SCC) (sequence-level diversity, i.e., fraction of unique categories explored over trajectory). Figure 3 summarizes results across Amazon, ML-1M, and Steam via reward–coverage trade-off across 100 steps. These results lead to following key observations:

- The closed-loop setting exposes a systematic reward–coverage tension that most baselines fail to address. As observed in results, baseline methods generally exhibit an inverse correlation between CR and SCC; pushing diversity typically hurts reward, while optimizing strictly for immediate relevance tends to collapse exposure into narrow bubbles.
- The unconstrained policy (Naive) and diversity-first interventions (NuGE) demonstrate distinct failure modes. Naive becomes trapped in content bubbles (lowest SCC), which ultimately caps its total utility. Conversely, NuGE promotes coverage at the cost of a severe collapse in utility (lowest CR), illustrating that indiscriminate exploration destroys relevance.
- Remaining methods like TD-VAE-CF, Filter, and RL remain strictly sub-optimal, reflecting systematic inefficiency. These methods fail to reach the performance frontier because their interventions are not coupled to an explicit saturation state; they either intervene too early (incurring unnecessary reward loss) or too late (failing to prevent confinement).
- CORAL achieves superior performance on both axes, outperforming Naive in Cumulative Reward while simultaneously maintaining high Coverage. By adaptively penalizing confinement, CORAL avoids the saturation traps that limit Naive's long-term utility. This indicates that breaking echo chambers is not merely a trade-off but a requisite for maximizing long-horizon reward, as it allows the policy to discover high-value regions that remain inaccessible to the Naive approach.

### 5.2.3. ROBUSTNESS: SIGNAL VALIDITY AND EXTREME FEEDBACK

Having established that CORAL's superiority across metrics on real-world datasets as well as under closed-loop dynamics, it remains to be seen whether its gains are robust in two complementary ways: (i) *modeling robustness*: whether the Hawkes intensity state $\lambda(t)$ is behaviorally meaningful as a proxy for exposure saturation, and (ii) *environment robustness* whether CORAL continues to stabilize the system

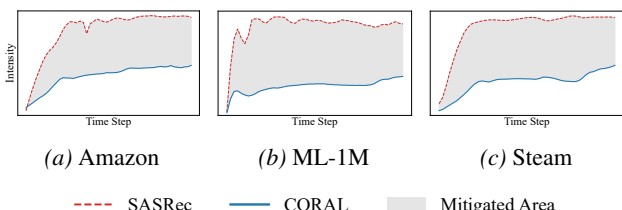

|        |        |        |
| *(a)* Amazon | *(b)* ML-1M | *(c)* Steam |

---- SASRec    —— CORAL    ▨ Mitigated Area

*Figure 4.* Evolution of echo chamber intensity across three datasets. The shaded regions demonstrate that CORAL significantly reduces Hawkes intensity compared to SASRec over time.

under adversarial, echo-chamber-amplifying user behaviors.

*Table 3.* Logistic regression analysis predicting user rejection ($y = 0$) based on Hawkes intensity across three datasets. Standard errors are in parentheses. The statistically significant positive $\beta_1$ confirms that higher Hawkes intensity is predictive of user fatigue.

| Dataset | Variable | Coef. ($\beta$) (S.E.) | $P$-value |
|---|---|---|---|
| **Amazon** | Constant ($\beta_0$) | -4.31 (0.21) | $< 0.001$ |
|            | Intensity ($\beta_1$) | 7.15 (0.64) | $< 0.001$ |
| **ML-1M**  | Constant ($\beta_0$) | -4.55 (0.14) | $< 0.001$ |
|            | Intensity ($\beta_1$) | 4.82 (0.55) | $< 0.001$ |
| **Steam**  | Constant ($\beta_0$) | -4.51 (0.29) | $< 0.001$ |
|            | Intensity ($\beta_1$) | 11.41 (0.75) | $< 0.001$ |

**Modeling robustness.** We validate whether the estimated Hawkes intensity $\lambda(t)$ tracks user saturation in a way that predicts disengagement. We implement a closed-loop simulation in which a Hawkes process interacts with a black-box LLM-based user simulator and a SASRec recommender. The simulation mimics cold-start: the LLM is initialized with five historical items derived from the history. To test generality across temporal dynamics, we simulate two user profiles: fatigue-prone (rapid interest decay) and obsessive (stable interest). The environment runs for 100 interaction steps, with Hawkes parameters updated every 10 steps via mini-batch online learning. We postulate that the estimated Hawkes intensity $\lambda(t)$ in Equation (2) serves as a proxy for user exposure saturation (without intervention). To validate this, we model the probability of a rejection event $y_t = 0$ at step $t$ using logistic regression:

$$P(y_t = 0 \mid \mathcal{H}_t) = \sigma(\beta_0 + \beta_1 \cdot \lambda(t)) \qquad (14)$$

where $\mathcal{H}_t$ denotes the interaction history up to time $t$ and $\sigma(\cdot)$ is the sigmoid function. Here, $y_t = 0$ denotes a rejection (e.g., skip/dislike). Under our simulator (prompt in Section C.5 in Appendix), rejections increase when recommendations become overly repetitive, so a higher rejection probability serves as a behavioral indicator of saturation. We then fit this logistic model, where a positive $\beta_1$ implies that a higher $\lambda(t)$ correlates with a higher rejection probability. Table 3 shows a consistently positive and significant

*Table 4.* Effect of category granularity on CORAL performance in closed-loop simulation. $|\mathcal{C}|$ denotes the number of categories.

| Dataset | Gran. | Method | $|\mathcal{C}|$ | CR↑ | SCC↑ |
|---|---|---|---|---|---|
| Amazon | Coarse | Naive | 28 | 0.231 | 0.236 |
| | | CORAL | 28 | 0.258 | 0.344 |
| | Fine | Naive | 112 | 0.213 | 0.179 |
| | | CORAL | 112 | 0.241 | 0.292 |
| ML-1M | Coarse | Naive | 18 | 0.270 | 0.111 |
| | | CORAL | 18 | 0.288 | 0.208 |
| | Fine | Naive | 38 | 0.271 | 0.097 |
| | | CORAL | 38 | 0.289 | 0.176 |
| Steam | Coarse | Naive | 10 | 0.531 | 0.186 |
| | | CORAL | 10 | 0.556 | 0.393 |
| | Fine | Naive | 44 | 0.525 | 0.167 |
| | | CORAL | 44 | 0.549 | 0.301 |

*Table 5.* Robustness to held-out category structure across three temporal phases. Naive is the unregulated baseline.

| Cumulative Reward (CR) ↑ | | | | |
|---|---|---|---|---|
| Dataset | Method | $t = 1\sim33$ | $t = 34\sim66$ | $t = 67\sim100$ |
| ML-1M | Naive | 0.328 | 0.259 | 0.242 |
| | CORAL | 0.194 | 0.245 | 0.283 |
| Amazon | Naive | 0.259 | 0.227 | 0.215 |
| | CORAL | 0.204 | 0.242 | 0.247 |
| Steam | Naive | 0.539 | 0.519 | 0.503 |
| | CORAL | 0.472 | 0.494 | 0.520 |
| Sequence Category Coverage (SCC) ↑ | | | | |
| Dataset | Method | $t = 1\sim33$ | $t = 34\sim66$ | $t = 67\sim100$ |
| ML-1M | Naive | 0.118 | 0.111 | 0.106 |
| | CORAL | 0.241 | 0.218 | 0.207 |
| Amazon | Naive | 0.248 | 0.237 | 0.224 |
| | CORAL | 0.381 | 0.320 | 0.332 |
| Steam | Naive | 0.176 | 0.167 | 0.161 |
| | CORAL | 0.340 | 0.278 | 0.277 |

intensity coefficient across Amazon, ML-1M, and Steam ($p < 0.001$), supporting the use of $\lambda(t)$ as an operational saturation signal for regulation.

**Environment robustness.** We next test the controller in an adverse closed-loop environment where users exhibit strong echo-chamber susceptibility, i.e., they preferentially select items similar to their consumption history, intentionally amplifying self-reinforcing feedback loops (prompt in Section C.5 in Appendix). Figure 4 shows the evolution of intensity over time. Under this amplified feedback, the SASRec baseline rapidly escalates toward a high-intensity saturated regime, whereas CORAL maintains a markedly flatter trajectory across Amazon, ML-1M, and Steam. The persistent gap over the interaction horizon indicates that the mitigation effect is sustained rather than transient, suggesting CORAL remains stable even when the environment actively reinforces homophily.

5.2.4. ROBUSTNESS TO CATEGORY STRUCTURE

We further evaluate whether CORAL depends strongly on a fixed category taxonomy. Table 4 compares coarse and fine category partitions in the closed-loop simulation. CORAL consistently improves over Naive in both cumulative reward (CR) and sequence category coverage (SCC) across all datasets. This indicates that the proposed risk-aware controller remains effective even when the category space becomes more fine-grained and the regulation problem becomes more fragmented. Table 5 evaluates robustness to previously unseen category structure by holding out a mid-frequency category during Hawkes training and measuring performance across three temporal phases. CORAL maintains substantially higher SCC than Naive across all datasets and phases. Although CORAL can incur a short-term CR cost in the first phase, it recovers over time and becomes competitive or better in later phases, suggesting that the con-

troller adapts as feedback from the new category structure is observed.

# 6. Conclusion

This paper investigates the challenge of maintaining high utility under endogenous feedback while preventing feedback-driven exposure concentration that traps users in echo chambers and collapses effective coverage over time. It proposes a novel framework, CORAL, that regulates exposure dynamics by estimating a history-dependent saturation state and penalizing actions via an upper-confidence risk bound on saturation violations. Theoretical analysis and empirical results supports effectiveness of the mechanism by establishing stability and efficient learning behavior. We leave the question of tighter confidence bounds and more robust state estimation for future work. Overall, this work provides a practical plug-in layer that can be applied atop different recommender backbones in real deployments.

# Impact Statement

Our framework mitigates feedback-driven exposure concentration in recommender systems by regulating exposure dynamics in the presence of endogenous user interactions. This can improve the robustness and transparency of large-scale personalization on user-centric platforms, potentially reducing echo-chamber effects in applications such as e-commerce, media streaming, and content discovery.

# Acknowledgments

This work is partially supported by the Australian Research Council (ARC) Under Grants DP220103717 and

LE220100078, and the National Natural Science Foundation of China under Grants No.62072257.

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

# Index

## A. Related Works

Time-ordered recommendation has progressed from early sequence models to modern deep sequential backbones such as GRU4Rec (Hidasi et al., 2015), SASRec (Kang & McAuley, 2018), and BERT4Rec (Sun et al., 2019). In deployment, these systems operate in a closed loop: recommendations influence user actions, which then shape future training and evaluation signals. Empirical and theoretical work shows that such feedback can narrow content diversity over time and induce self-reinforcing dynamics (i.e., filter-bubble/echo-chamber effects) (Nguyen et al., 2014; Jiang et al., 2019). Contemporary studies have advanced this understanding by employing LLM-based user agents to realistically simulate the formation of filter bubbles (Zhang et al., 2024; Sukiennik et al., 2025). A large line of mitigation methods addresses concentration via post-hoc diversification and re-ranking (e.g., topic diversification or MMR-style objectives) (Ziegler et al., 2005; Carbonell & Goldstein, 1998). While effective at improving diversity metrics, these approaches are typically not designed as a reliability layer that explicitly regulates a history-dependent saturation state under endogenous feedback. Another area of research studies safe learning and constrained decision-making, balancing exploration against constraint violations (Auer et al., 2002; Achiam et al., 2017; Badanidiyuru et al., 2018). Separately, temporal point processes have been used to model self-exciting user activity and non-stationary interaction dynamics (Hawkes, 1971; Du et al., 2016). These point-process formulations primarily serve as predictive models of event arrivals, and are typically not coupled to explicit constraint enforcement in sequential decision-making.

## B. Notation Table

We summarize the key notations used throughout the paper in Table 6.

## C. Detailed Experimentation Details

In the main paper, we introduced multiple datasets and a closed-loop evaluation protocol to assess CORAL under feedback-driven recommendation. Below, we provide further details on datasets and preprocessing, sampling and splitting, the closed-loop interaction protocol and click feed-

back model, periodic backbone updates, base recommendation models, and baseline intervention layers used for comparison.

### C.1. Implement Details

We elaborate on the implementation details of the experiments conducted. We conduct experiments on an NVIDIA RTX A5000 GPU. To ensure fair comparison, we adopt consistent hyperparameters across all methods: hidden embedding dimension of 100, batch size of 32, and L2-normalized parameter initialization. Specifically, we adopt the same batch size, embedding dimension, learning rate, and other training configurations as the original implementations of each baseline model.

For the backbone models ( SASRec and Bert4Rec), we follow the original architectures to ensure the validity of the sequential patterns learned. Specifically, we set the number of self-attention layers and attention heads to 2 (i.e., n_layers=2, n_heads=2). These backbones are pretrained with a learning rate of $1 \times 10^{-3}$ (lr=0.001) using the Adam optimizer before being integrated into the CORAL framework or baseline policies.

Regarding the specific hyperparameters, we fix the tolerance factor $\gamma$ and the threshold $\tau$ at 2 (i.e., $\gamma$=2, $\tau$=2) across all datasets to maintain stability in exposure saturation estimation for rating-based offline histories. For the remaining critical hyperparameters, we conduct a grid search and sensitivity analysis to identify the optimal settings. Specifically, we tune the exploration coefficient $\kappa$, the safety limit $\lambda_{max}$, and the risk budget $\delta$ from the following ranges: Exploration coefficient $\kappa \in \{0.5, 1.0, 2.0, 3.0\}$; Safety limit $\lambda_{max} \in \{0.3, 0.5, 0.7, 0.9, 1.1\}$; Risk budget $\delta \in \{0.01, 0.05, 0.1, 0.2\}$ We report the performance based on the configuration that achieves the best trade-off between recommendation utility and exposure health metrics on the validation set.

### C.2. Datasets

- **ML-1M** (Harper & Konstan, 2015)[2]: movie ratings converted to implicit sequential feedback.

- **Steam** (Kang & McAuley, 2018)[3]: user–game interaction data collected from the Steam platform, where purchase and play records are treated as implicit sequential feedback.

- **Amazon** (He & McAuley, 2016)[4]: the Amazon dataset

[2]https://grouplens.org/datasets/movielens/1m/
[3]https://cseweb.ucsd.edu/~jmcauley/datasets.html
[4]http://jmcauley.ucsd.edu/data/amazon/

containing mixed category product information, obtained by filtering the last 2M interactions chronologically.

All datasets are time-ordered. We apply a $k$-core filtering strategy (with $k = 5$ by default) and standard preprocessing for sequential recommendation, following prior work (Kang & McAuley, 2018; Sun et al., 2019). Item categories are obtained from the dataset metadata. The statistics of the datasets have been summarized in Table 7.

### C.3. Closed-Loop Interaction Protocol

We simulate a closed-loop recommendation trajectory over a horizon $T$. At step $t$, the backbone computes scores $s_t(a)$ for candidate items. The policy (CORAL or baseline) selects an action $a_t$. Unlike static environments, the user feedback $y_t \in \{0, 1\}$ is generated dynamically by an LLM-based user simulator instantiated with specific personas and interaction histories. The resulting interaction tuple $(a_t, y_t)$ is logged to the replay buffer $\mathcal{D}$ for periodic user history and CORAL updates. This setup allows us to observe the long-term impact of intervention strategies on user utility and system diversity.

### C.4. Category Granularity Robustness Protocol

To assess CORAL's robustness to different levels of category taxonomy granularity, we construct two independent category partitions for each dataset. The construction differs by dataset. For Amazon and Steam, we start from top-level metadata labels as the *coarse* taxonomy and expand hierarchically to obtain the *fine* taxonomy. For Amazon, we move from broad product types to leaf-level subcategories and for Steam, from top-level genre groups to second-level genre tags. For ML-1M, the original dataset already provides fine-grained multi-genre combination labels (e.g., *Comedy—Drama*, *Action—Adventure—Thriller* each treated as a distinct category, giving 38 unique tags), so the *fine* taxonomy corresponds to these original labels, while the *coarse* taxonomy is derived by collapsing combinations into their primary genre (e.g., any tag containing *Action* maps to the *Action* group), resulting in 18 broader categories. The resulting category counts are reported in the $|\mathcal{C}|$ column of Table 4. For each dataset and taxonomy level, we run an independent closed-loop simulation following the base protocol in Section C.3. The backbone (SASRec or Bert4Rec) is pretrained on the training split, category assignments are replaced with those of the target taxonomy before simulation, and the LLM-based user simulator (Section C.5) is initialized with five items from the user's training history. CORAL's Hawkes intensity model is re-fitted using (Equation (4)) on the training split under the target taxonomy.

## C.5. Held-Out Category Robustness Protocol

To evaluate CORAL's ability to adapt to a completely new category unseen during training, we identify for each dataset a single *held-out* category. We exclude the top three most frequent categories to avoid distorting the training distribution, and exclude the bottom quartile to ensure sufficient occurrence in the test stream. Both the backbone model, either SASRec or Bert4Rec, and CORAL's Hawkes intensity model are trained after fully removing this held-out category. All interactions involving items from this category are excluded from the training split. Consequently, the backbone learns no item embeddings for the held-out category, while CORAL is calibrated only on the remaining categories and has no pre-initialized intensity estimate for the held-out category. This design is therefore equivalent to a cold-start scenario in which a new category is launched after the system has already been calibrated. During the closed-loop simulation, items from the held-out category are gradually introduced into the candidate pool. A small number of new items are added at each phase boundary, and both the backbone model and CORAL adapt incrementally through online updates as these items receive user feedback. To examine the temporal dynamics of this process, the 100-step horizon is divided into three approximately equal phases: an early phase covering the first 33 steps, when CORAL's estimates are still uninformative; an adaptation phase covering steps 34 to 66, when accumulated interactions allow the estimates to stabilize; and a mature phase covering steps 67 to 100, when the risk-aware policy has adapted to the new category structure.

**LLM-based User Simulator**  To simulate realistic user behavior, we employ the Gemma-3-12B-IT large language model (Team et al., 2025) as a user simulator, following prior work (Feng et al., 2025). The simulator is initialized with a user profile containing a ground-truth interaction history $\mathcal{H}_{t-1}$ and preferred categories. We design two distinct prompting strategies to evaluate policy robustness under different user behaviors:

1. **Standard User:** Acts based on intrinsic interests and personality traits (e.g., prone to boredom or obsessive), evaluating recommendations against their general preferences.

2. **Echo Chamber User:** Simulates a user with high confirmation bias. We inject a specific constraint (highlighted in red below) requiring the user to strictly prefer items similar to their history, thereby mimicking feedback loop dynamics.

The prompts governing these behaviors are detailed below. The model outputs an integer decision: the item ID (interpreted as a click, $y_t = 1$) or 0 (rejection, $y_t = 0$).

---

**Standard Evaluation Prompt**

**Role**: [domain] User simulator.

**Context**: You have a history of [history_str] and prefer categories [cate_str]. You [get bored very quickly / have obsessive personality].

**Task**: Evaluate the recommendation list [rec_str]. Select its ID [item_num] if you are interested. Select 0 if you reject it (due to boredom or mismatch).

**Output**: choose integer decision only (0 or [item_num]).

---

**Echo Chamber Evaluation Prompt**

**Role**: [domain] User simulator.

**Context**: You have a history of [history_str] and prefer categories [cate_str]. You [get bored very quickly / have obsessive personality], you strongly prefer items similar to your history.

**Task**: Evaluate the recommendation list [rec_str]. Select its ID [item_num] if you are interested. Select 0 if you reject it (due to boredom or mismatch).

**Output**: choose integer decision only (0 or [item_num]).

---

## C.6. Backbone Recommendation Models

We instantiate CORAL on two representative recommendation backbones:

- **SASRec** (Kang & McAuley, 2018): a unidirectional self-attention-based sequential recommender that models users' interaction histories for next-item prediction.

- **BERT4Rec** (Sun et al., 2019): a bidirectional Transformer-based sequential recommender trained with a Cloze-style masked item prediction objective.

Backbone hyperparameters are fixed across CORAL and all baselines within each dataset. Training uses Adam with early stopping on validation performance.

## C.7. Baseline Methods

(i) Filter (Antikacioglu & Ravi, 2017): A similarity-based safety mechanism that post-processes the recommendation list. It computes the cosine similarity between a candidate item and the mean-pooled representation of the user's interaction history. Candidates exceeding a predefined similarity threshold are removed to prevent excessive homogeneity.

(ii) TD-VAE-CF (Gao et al., 2022): A variational autoencoder-based collaborative filtering framework de-

signed for targeted diversification. It modifies the user's latent representation by shifting it along Concept Activation Vectors, which define directions in the latent space corresponding to specific item attributes or subtopics. This approach balances reconstruction accuracy with the diversification of exposure across different concept clusters.

(iii) NuGE (Wang et al., 2024): A graph-based responsible recommendation framework that acts as an intermediate agent between the base recommender and the user. It employs a Belief Filter Bubble Detection (FBDetect) module to identify users with extreme belief imbalances and a Belief Nudging module to generate new items inserted into history.

(iv) RL (Li et al., 2023): A reinforcement learning framework for controllable recommendation that addresses the trade-off between user utility and filter bubble mitigation. It models the recommendation process as a Markov Decision Process (MDP) where the agent (a policy/intervention layer) optimizes a reward function combining long-term user engagement and an exposure diversity metric. The method dynamically adjusts the exposure of items to ensure a balanced distribution across different groups or interests over time.

We include the rule-based filter as a widely used safety heuristic; additional rule variants behave similarly and are omitted for brevity.

## C.8. Evaluation Metrics

We evaluate performance using two sets of metrics corresponding to the static step-by-step offline and simulated online environments. Let $\mathcal{U}$ denote the set of users, $\mathcal{I}$ the set of items, and $\mathcal{C}$ the set of item categories.

**Static Offline Metrics.** For each user $u \in \mathcal{U}$, let $\mathcal{I}_u^+$ be the ground-truth positive item set, and $\hat{\mathcal{I}}_{u,K}$ be the ordered list of top-$K$ recommended items.

- **Recall@K and MRR@K**: Standard utility metrics.

- **Exposure Concentration (EC)**: To quantify the "collapse" of exposure onto a narrow subset of categories (echo chamber effect), we utilize the Gini coefficient of the category exposure distribution. Let $N(c) = \sum_{u \in \mathcal{U}} \sum_{i \in \hat{\mathcal{I}}_{u,K}} \mathbb{I}(\text{cat}(i) = c)$ be the global exposure count for category $c$.

$$EC = \frac{\sum_{c \in \mathcal{C}} \sum_{c' \in \mathcal{C}} |N(c) - N(c')|}{2|\mathcal{C}| \sum_{c \in \mathcal{C}} N(c)}. \quad (15)$$

A higher EC indicates greater inequality in category exposure.

- **Peak Exposure (PS)**: Let $D_{u,t}$ denote the exposure-load for user $u$ at step $t$. We define PS as the maximum saturation level observed across all users and time steps:

$$PS = \max_{u \in \mathcal{U}, t} D_{u,t}. \quad (16)$$

**Simulated Online Metrics.** In the closed-loop simulation over a horizon $T$, let $r_t \in \{0, 1\}$ be the reward at time $t$ and $c_t$ be the category of the recommended item.

- **Cumulative Reward (CR)**: Measures the long-term utility accumulation:

$$CR = \frac{1}{|\mathcal{U}| * T} \sum_{u \in \mathcal{U}} \sum_{t=1}^{T} r_{u,t}. \quad (17)$$

- **Sequence Category Coverage (SCC)**: Measures the diversity of the exploration process. Let $\mathcal{C}_{u,T} = \{c_{u,1}, \ldots, c_{u,T}\}$ be the sequence of categories exposed to user $u$. SCC is the fraction of unique categories covered:

$$SCC = \frac{1}{|\mathcal{U}|} \sum_{u \in \mathcal{U}} \frac{|\text{Unique}(\mathcal{C}_{u,T})|}{|\mathcal{C}|}. \quad (18)$$

## D. Additional Experiments

### D.1. Sensitivity Analysis

We study the sensitivity of CORAL to three key hyperparameters that directly govern the exploration–exploitation and safety–utility trade-offs. Figure 5 reports Coverage and Cumulative Reward as we vary the safety threshold $\lambda_{\max}$, exploration coefficient $\kappa$, and risk budget $\delta$ across Amazon, ML-1M, and Steam. The results lead to the following observations:

- *Safety threshold* ($\lambda_{\max}$) controls the primary safety–utility. Across all three datasets, decreasing $\lambda_{\max}$ leads to higher Coverage but lower Reward, while increasing $\lambda_{\max}$ relaxes the constraint and shifts the operating point toward higher Reward with reduced Coverage. This monotone trend is expected: tighter thresholds trigger regulation more frequently, promoting diversity at the cost of suppressing repeatedly high-utility exposure. Notably, as $\lambda_{\max}$ becomes sufficiently large, the safety constraint ceases to bind, and CORAL effectively reverts to the original unconstrained model state.

- *Exploration coefficient* ($\kappa$) exhibits a "too little vs. too much" exploration effect. Reward follows a consistent inverted-U pattern with respect to $\kappa$ as small $\kappa$ under-explores and can remain trapped in locally optimal but confined trajectories (lower Coverage), whereas overly large $\kappa$ over-explores and sacrifices relevance. Moderate exploration (around the mid-range setting in Figure 5) achieves the best Reward while maintaining strong Coverage, indicating that CORAL benefits from exploration that is substantial but not dominant.

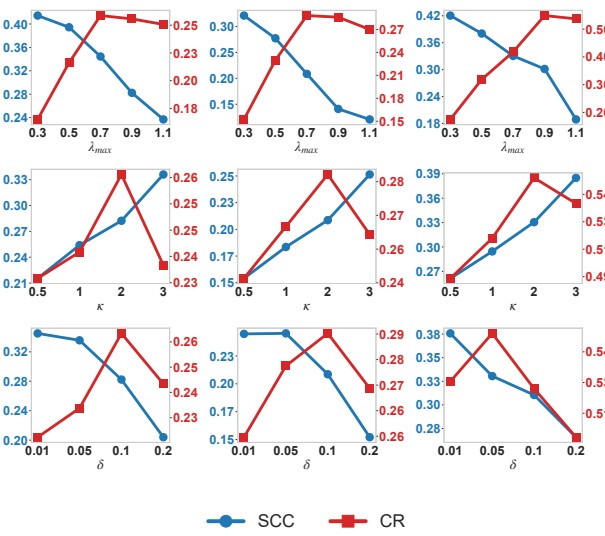

*Figure 5.* The results of hyper-parameter sensitivity analysis. The three columns from left to right correspond to the datasets Amazon, ML-1M, and Steam, respectively.

- *Risk budget ($\delta$)* tunes conservatism through the confidence bound. Performance is best at intermediate $\delta$ values. Tighter budgets make the controller overly conservative, increasing regulatory frequency and reducing Reward, while overly loose budgets weaken the risk signal and reduce coverage gains. The consistent peak at moderate $\delta$ across datasets suggests that effective regulation requires uncertainty control that is neither overly pessimistic nor overly permissive.

## D.2. Ablation Study

We isolate the contribution of each CORAL component by comparing the full model to four variants on Amazon, ML-1M, and Steam datasets. Specifically, we consider: (i) w/o Hawkes: replace the Hawkes intensity estimator with a simple moving average (SMA); (ii) w/o Risk: remove the variance-based uncertainty term from the risk estimate; (iii) w/o Adaptive: replace the soft penalty with a hard threshold; and (iv) w/o Explore: disable exploration by setting $\kappa = 0$. Table 8 reports the Coverage and Cumulative Reward. We make the following observation:

- The full model consistently exceeds all ablations, indicating that the gains are not attributed to a single component. Across all three datasets, CORAL achieves the best joint outcome in Coverage and Reward (e.g., on Steam: Coverage= 0.3015 and Reward= 0.5490 in Table 8).
- Hawkes-based temporal modeling is necessary for responsive regulation under non-stationary interest. Replacing Hawkes with SMA (w/o Hawkes) produces a

consistent drop in both Coverage and Reward across datasets. This validates the role of Hawkes as, without a temporally aware intensity with decay and burst sensitivity, the controller reacts slowly to changes, which can over-penalize recently saturated categories and fail to admit them when interest shifts.

- Risk-awareness and adaptive penalization jointly prevent over-conservative or unstable control. Both removing uncertainty (w/o Risk) and replacing the soft penalty with a hard threshold (w/o Adaptive) degrade performance on both metrics. Point-estimate control and hard switching make regulation either overly conservative or overly brittle, which simultaneously limits safe exploration (hurting Coverage) and suppresses legitimate high-reward recommendations (hurting Reward).
- Exploration is the necessary for coverage gains. Disabling exploration (w/o Explore, $\kappa = 0$) causes the largest Coverage drop across datasets, while Reward becomes less robust. This is consistent with greedy exploitation. Once regulation constrains the action set, the policy repeatedly selects the locally best remaining category, preventing "bubble escape" even when additional safe alternatives exist.

# E. Proofs

## E.1. Proof for Lemma 1

**Proof.** We aim to verify the defining properties of a martingale difference sequence (MDS) for the sequence $\{M_{t+1}\}_{t \geq 1}$ with respect to the filtration $\{\mathcal{G}_t\}_{t \geq 1}$. By definition, we have

$$L_{t+1} = \mathbb{I}(D_{t+1} > \lambda_{\max}) \in \{0, 1\},$$

hence $L_{t+1}$ is integrable and $\mathbb{E}[|L_{t+1}|] \leq 1$. Since conditional expectations of integrable random variables are integrable, by Jensen's inequality, we can say: $R_t(c(a_t)) = \mathbb{E}[L_{t+1} \mid \mathcal{G}_t]$ is integrable as well. Therefore,

$$
\begin{aligned}
\mathbb{E}[|M_{t+1}|] &= \mathbb{E}\big[\big|L_{t+1} - R_t(c(a_t))\big|\big] \\
&\leq \mathbb{E}[|L_{t+1}|] + \mathbb{E}[|R_t(c(a_t))|] < \infty.
\end{aligned}
\tag{i}
$$

By construction, $\mathcal{G}_t = \sigma(\mathcal{H}_{t-1}, a_t)$ is the sigma-algebra containing all randomness revealed up to decision time $t$. The random variable $R_t(c(a_t)) = \mathbb{E}[L_{t+1} \mid \mathcal{G}_t]$ is $\mathcal{G}_t$-measurable by definition. Moreover, $L_{t+1}$ is determined once the feedback at time $t$ is realized, so it is $\sigma(\mathcal{H}_t)$-measurable. Since $\sigma(\mathcal{H}_t) \subseteq \mathcal{G}_{t+1}$, it follows that $L_{t+1}$ is $\mathcal{G}_{t+1}$-measurable. Therefore, the difference is also measurable, i.e.;

$$M_{t+1} = L_{t+1} - R_t(c(a_t)) \quad \text{is } \mathcal{G}_{t+1}\text{-measurable.} \tag{ii}$$

Now given the definition we have: $R_t(c(a_t)) = \mathbb{E}[L_{t+1} \mid \mathcal{G}_t]$, we compute the conditional expectation of $M_{t+1}$ given $\mathcal{G}_t$:

$$\mathbb{E}[M_{t+1} \mid \mathcal{G}_t] = \mathbb{E}[L_{t+1} - R_t(c(a_t)) \mid \mathcal{G}_t]$$
$$= \mathbb{E}[L_{t+1} \mid \mathcal{G}_t] - \mathbb{E}[R_t(c(a_t)) \mid \mathcal{G}_t]. \quad \text{(iii)}$$

Since $R_t(c(a_t))$ is $\mathcal{G}_t$-measurable (2), we have $\mathbb{E}[R_t(c(a_t)) \mid \mathcal{G}_t] = R_t(c(a_t))$. Substituting into (iii) and using the earlier property on $R_t$ gives us:

$$\mathbb{E}[M_{t+1} \mid \mathcal{G}_t] = \mathbb{E}[L_{t+1} \mid \mathcal{G}_t] - R_t(c(a_t)) = 0. \quad \text{(iv)}$$

This holds for every $t \geq 1$, hence $\{M_{t+1}\}_{t \geq 1}$ is a martingale difference sequence with respect to $\{\mathcal{G}_t\}_{t \geq 1}$.

Hence Proved. $\qquad \square$

### E.2. Proof for Theorem 1

**Proof.** We try to control the upper deviation of the latent violation risk $\bar{R}_t(c)$ from its empirical estimate $\hat{R}_t(c)$ uniformly over all $t \leq T$ and all $c \in \mathcal{C}$.

Given any category $c \in \mathcal{C}$ and any time $t \in \{1, \dots, T\}$, let:

$$N := N_t(c) = \sum_{k=1}^{t-1} \mathbb{I}(c(a_k) = c). \quad \text{(i)}$$

If $N = 0$, then by definition $\hat{R}_t(c) = 0$ and $\bar{R}_t(c) = 0$, so (9) holds trivially. In what follows, assume $N \geq 1$, then we can write:

$$\hat{R}_t(c) = \frac{1}{N} \sum_{k=1}^{t-1} \mathbb{I}(c(a_k) = c) \, L_{k+1},$$
$$\bar{R}_t(c) = \frac{1}{N} \sum_{k=1}^{t-1} \mathbb{I}(c(a_k) = c) \, \mathbb{E}[L_{k+1} \mid \mathcal{G}_k], \quad \text{(ii)}$$

where $\mathcal{G}_k = \sigma(\mathcal{H}_{k-1}, a_k)$.

For simplicity, for each $k \leq t - 1$, let us define $p_k := \mathbb{E}[L_{k+1} \mid \mathcal{G}_k] \in [0, 1]$, such that:

$$Z_k := \mathbb{I}(c(a_k) = c) \, (L_{k+1} - p_k). \quad \text{(iii)}$$

Using $p_k$, we have

$$\mathbb{E}[Z_k \mid \mathcal{G}_k] = \mathbb{I}(c(a_k) = c) \, (\mathbb{E}[L_{k+1} \mid \mathcal{G}_k] - p_k) = 0. \quad \text{(iv)}$$

Moreover, because $L_{k+1} \in \{0, 1\}$ and $p_k \in [0, 1]$, we have the bounded increment property:

$$|Z_k| \leq 1. \quad \text{(v)}$$

Defining the partial sum:

$$S_t := \sum_{k=1}^{t-1} Z_k. \quad \text{(vi)}$$

Then $\{S_t\}$ is a martingale with respect to the filtration $\{\mathcal{G}_k\}$, and by (ii)–(iii) we obtain:

$$\hat{R}_t(c) - \bar{R}_t(c) = \frac{1}{N} \sum_{k=1}^{t-1} \mathbb{I}(c(a_k) = c) \, (L_{k+1} - p_k) = \frac{S_t}{N}. \quad \text{(vii)}$$

We now utilize Freedman's inequality for martingales with bounded differences. Since $|Z_k| \leq 1$, the deviation of $S_t$ is controlled by its cumulative conditional variance $V_t := \sum_{k=1}^{t-1} \mathbb{E}[Z_k^2 \mid \mathcal{G}_k]$. Specifically, for any $x > 0$, Freedman's inequality guarantees:

$$\mathbb{P}\left(S_t \geq \sqrt{2V_t x} + \frac{x}{3}\right) \leq e^{-x} \quad \text{and}$$
$$\mathbb{P}\left(-S_t \geq \sqrt{2V_t x} + \frac{x}{3}\right) \leq e^{-x}. \quad \text{(viii)}$$

Given this, we now upper bound $V_t$ in terms of $\bar{R}_t(c)$. Since $\mathbb{I}(c(a_k) = c)^2 = \mathbb{I}(c(a_k) = c)$ and $Z_k = \mathbb{I}(c(a_k) = c)(L_{k+1} - p_k)$, we have:

$$\mathbb{E}[Z_k^2 \mid \mathcal{G}_k] = \mathbb{I}(c(a_k) = c) \, \mathbb{E}[(L_{k+1} - p_k)^2 \mid \mathcal{G}_k]$$
$$= \mathbb{I}(c(a_k) = c) \, \mathrm{Var}(L_{k+1} \mid \mathcal{G}_k)$$
$$= \mathbb{I}(c(a_k) = c) \, p_k(1 - p_k). \quad \text{(ix)}$$

Therefore:

$$V_t = \sum_{k=1}^{t-1} \mathbb{I}(c(a_k) = c) \, p_k(1 - p_k). \quad \text{(x)}$$

Let $\bar{p} := \bar{R}_t(c) = \frac{1}{N} \sum_{k=1}^{t-1} \mathbb{I}(c(a_k) = c) \, p_k$. The function $f(p) = p(1 - p)$ is concave on $[0, 1]$, hence by Jensen's inequality,

$$\frac{1}{N} \sum_{k=1}^{t-1} \mathbb{I}(c(a_k) = c) \, p_k(1 - p_k) \leq \bar{p}(1 - \bar{p}). \quad \text{(xi)}$$

Combining (x) and (xi) gives us:

$$V_t \leq N \, \bar{R}_t(c) \big(1 - \bar{R}_t(c)\big). \quad \text{(xii)}$$

Substitute (xii) into the second inequality of (viii) and divide by $N$. Using (vii), we obtain: for any $x > 0$:

$$\mathbb{P}\left(\bar{R}_t(c) - \hat{R}_t(c) \geq \sqrt{\frac{2\,\bar{R}_t(c)\big(1 - \bar{R}_t(c)\big)\,x}{N}} + \frac{x}{3N}\right) \leq e^{-x}. \quad \text{(xiii)}$$

For the sake of simplicity of notations, let $d := \bar{R}_t(c) - \hat{R}_t(c)$. On the event in (xiii) we have $d \geq 0$. Also let $\hat{v} := \hat{R}_t(c)\big(1 - \hat{R}_t(c)\big)$ and $a := x/N$. Then (xiii) is equivalent to:

$$d \leq \sqrt{2\,\bar{R}_t(c)\big(1 - \bar{R}_t(c)\big)\,a} + \frac{a}{3}. \quad \text{(xiv)}$$

We now relate $\bar{R}_t(c)\big(1 - \bar{R}_t(c)\big)$ to $\hat{v}$. Let $f(p) = p(1-p)$. Since $f'(p) = 1 - 2p$ and $|f'(p)| \leq 1$ on $[0,1]$, the simple mean value theorem gives the following Lipschitz bound:

$$|f(p) - f(q)| \leq |p - q| \quad \text{for all } p, q \in [0, 1]. \qquad \text{(xv)}$$

Applying (xv) with $p = \bar{R}_t(c)$ and $q = \hat{R}_t(c)$ gives

$$\bar{R}_t(c)\big(1 - \bar{R}_t(c)\big) \leq \hat{R}_t(c)\big(1 - \hat{R}_t(c)\big) + |\bar{R}_t(c) - \hat{R}_t(c)| = \hat{v} + d. \qquad \text{(xvi)}$$

Substituting (xvi) into (xiv) gives us:

$$d \leq \sqrt{2(\hat{v} + d)a} + \frac{a}{3}. \qquad \text{(xvii)}$$

We now solve (xvii) for $d$. Let $b := a/3$. Since $d \geq 0$, we have $(d - b)_+ \leq d$, and from (xvii) it follows that $d - b \leq \sqrt{2(\hat{v} + d)a}$ whenever $d \geq b$; if $d < b$, the final bound holds trivially. Assume $d \geq b$ and square both sides. We get:

$$(d - b)^2 \leq 2a(\hat{v} + d) = 2a\hat{v} + 2ad. \qquad \text{(xviii)}$$

Expanding and rearranging terms gives us:

$$d^2 - (2b + 2a)d + (b^2 - 2a\hat{v}) \leq 0. \qquad \text{(xix)}$$

The largest root of the quadratic on the left-hand side upper bounds $d$, hence,

$$d \leq b + a + \sqrt{a^2 + 2ab + 2a\hat{v}}. \qquad \text{(xx)}$$

Using $\sqrt{u + v} \leq \sqrt{u} + \sqrt{v}$ with $u = 2a\hat{v}$ and $v = a^2 + 2ab$, we obtain:

$$\sqrt{a^2 + 2ab + 2a\hat{v}} \leq \sqrt{2a\hat{v}} + \sqrt{a^2 + 2ab}. \qquad \text{(xxi)}$$

Since $b = a/3$, we have $a^2 + 2ab = a^2(1 + 2/3) = \frac{5}{3}a^2$, hence $\sqrt{a^2 + 2ab} = \sqrt{5/3}\, a \leq 1.3\, a$. Substituting into (xx) results in the simpler bound:

$$d \leq \sqrt{2a\hat{v}} + (b + a + 1.3a) \leq \sqrt{2a\hat{v}} + 3a, \qquad \text{(xxii)}$$

where we used $b = a/3$ and $b + a + 1.3a = 2.633a \leq 3a$.

Now, we recall $a = x/N$ and $\hat{v} = \hat{R}_t(c)(1 - \hat{R}_t(c))$, inequality (xxii) becomes

$$\bar{R}_t(c) - \hat{R}_t(c) \leq \sqrt{\frac{2\,\hat{R}_t(c)\big(1 - \hat{R}_t(c)\big)\, x}{N}} + \frac{3x}{N}. \quad \text{(xxiii)}$$

To ensure the bound holds simultaneously for all $t \in \{1, \ldots, T\}$ and all categories $c \in \mathcal{C}$ with probability at least $1 - \delta$, we apply a union bound. There are at most $T \times |\mathcal{C}|$ such events. For any fixed pair $(t, c)$, the failure probability is at most $2e^{-x}$. Summing this risk over all steps and categories, we require the total failure probability to be at most $\delta$:

$$\sum_{t=1}^{T} \sum_{c \in \mathcal{C}} 2e^{-x} = 2T|\mathcal{C}|e^{-x} \leq \delta. \qquad \text{(xxiv)}$$

We solve for $x$ by isolating the exponential term and taking the natural logarithm:

$$e^{-x} \leq \frac{\delta}{2T|\mathcal{C}|},$$

$$-x \leq \ln\left(\frac{\delta}{2T|\mathcal{C}|}\right),$$

$$x \geq -\ln\left(\frac{\delta}{2T|\mathcal{C}|}\right) = \ln\left(\frac{2T|\mathcal{C}|}{\delta}\right).$$

We set $x$ to this limiting value: $x := \ln(2T|\mathcal{C}|/\delta)$. Substituting $N = N_t(c)$ and this value of $x$ into (xxiii) gives us:

$$\bar{R}_t(c) \leq \hat{R}_t(c) + \sqrt{\frac{2\,\hat{R}_t(c)\big(1 - \hat{R}_t(c)\big) \ln\left(\frac{2T|\mathcal{C}|}{\delta}\right)}{N_t(c)}}$$

$$+ \frac{3\ln\left(\frac{2T|\mathcal{C}|}{\delta}\right)}{N_t(c)}.$$

Hence Proved. $\qquad \square$

### E.3. Proof of Theorem 2

**Proof.** Given a time $t_0$, we have $D_{t_0} \geq \lambda_{\max} - \epsilon$ for some $\epsilon > 0$. Let

$$L := \lambda_{\max} - \epsilon, \qquad a := e^{-\beta_{\min}}, \qquad q := \frac{\alpha_{\text{safe}}}{1 - a}.$$

Since $\beta_{\min} > 0$, we have $0 < a < 1$. By assumption, during regulation, the selected category induces residual aggregate excitation bounded by $\alpha_{\text{safe}}$ after Hawkes decay. Hence, for all regulation steps,

$$D_{t+1} \leq aD_t + \alpha_{\text{safe}}. \qquad \text{(i)}$$

Iterating (i) from $t_0$ gives us, for any integer $\tau \geq 0$:

$$D_{t_0 + \tau} \leq a^\tau D_{t_0} + \alpha_{\text{safe}} \sum_{j=0}^{\tau-1} a^j$$

$$= a^\tau D_{t_0} + \alpha_{\text{safe}} \frac{1 - a^\tau}{1 - a}. \qquad \text{(ii)}$$

Equivalently, using $q = \frac{\alpha_{\text{safe}}}{1-a}$, we can write:

$$D_{t_0 + \tau} \leq a^\tau(D_{t_0} - q) + q. \qquad \text{(iii)}$$

Now, by the condition in Theorem 2, we have

$$\alpha_{\text{safe}} < (1 - e^{-\beta_{\min}})(\lambda_{\max} - \epsilon). \qquad \text{(iv)}$$

Using $a = e^{-\beta_{\min}}$ and $L = \lambda_{\max} - \epsilon$, (iv) implies:

$$q = \frac{\alpha_{\text{safe}}}{1-a} < L. \qquad \text{(v)}$$

Now by the definition of $\tau_{\text{reg}}$, we find the smallest integer $\tau$ such that $D_{t_0+\tau} \leq L$. Using (iii), it suffices to require that:

$$a^\tau(D_{t_0} - q) + q \leq L. \qquad \text{(vi)}$$

Since $D_{t_0} \geq L$ and $q < L$, rearranging (vi) gives us:

$$a^\tau \leq \frac{L-q}{D_{t_0}-q}. \qquad \text{(vii)}$$

Using $a = e^{-\beta_{\min}}$, (vii) is satisfied whenever:

$$\tau \geq \frac{1}{\beta_{\min}} \ln\left(\frac{D_{t_0}-q}{L-q}\right). \qquad \text{(viii)}$$

Substituting back $L = \lambda_{\max} - \epsilon$ and $q = \frac{\alpha_{\text{safe}}}{1-e^{-\beta_{\min}}}$, (viii) shows that any integer

$$\tau \geq \left\lceil \frac{1}{\beta_{\min}} \ln\left(\frac{D_{t_0} - \frac{\alpha_{\text{safe}}}{1-e^{-\beta_{\min}}}}{(\lambda_{\max} - \epsilon) - \frac{\alpha_{\text{safe}}}{1-e^{-\beta_{\min}}}}\right) \right\rceil$$

guarantees $D_{t_0+\tau} \leq L$. Therefore, by the definition of $\tau_{\text{reg}}$ as the first regulation step reaching the safe region, we obtain:

$$\tau_{\text{reg}} \leq \left\lceil \frac{1}{\beta_{\min}} \ln\left(\frac{D_{t_0} - \frac{\alpha_{\text{safe}}}{1-e^{-\beta_{\min}}}}{(\lambda_{\max} - \epsilon) - \frac{\alpha_{\text{safe}}}{1-e^{-\beta_{\min}}}}\right) \right\rceil. \qquad \text{(ix)}$$

Hence Proved. $\qquad \square$

### E.4. Proof of Corollary 1

**Proof.** Under the neutral-category special case, the selected regulation category has zero residual aggregate excitation, i.e.,

$$\alpha_{\text{safe}} = 0. \qquad \text{(i)}$$

Substituting (i) into Theorem 2 gives us:

$$\tau_{\text{reg}} \leq \left\lceil \frac{1}{\beta_{\min}} \ln\left(\frac{D_{t_0}}{\lambda_{\max} - \epsilon}\right) \right\rceil. \qquad \text{(ii)}$$

Hence Proved. $\qquad \square$

### E.5. Proof for Theorem 3

**Proof.** We begin by decomposing regret into (i) regret incurred during regulation rounds and (ii) regret incurred during non-regulation rounds.

Let $\mathcal{T}_{\text{reg}} \subseteq \{1, \ldots, T\}$ denote the set of regulation rounds and $\mathcal{T}_{\text{nor}} := \{1, \ldots, T\} \setminus \mathcal{T}_{\text{reg}}$ the remaining rounds. Then

we can write:

$$\begin{aligned} \mathfrak{R}(T) &= \sum_{t=1}^{T} (\mu_{c^\star} - \mu_{c_t}) \\ &= \sum_{t \in \mathcal{T}_{\text{nor}}} (\mu_{c^\star} - \mu_{c_t}) + \sum_{t \in \mathcal{T}_{\text{reg}}} (\mu_{c^\star} - \mu_{c_t}). \end{aligned} \qquad \text{(i)}$$

By definition, $\Delta_{\max} := \max_{c \in \mathcal{C}}(\mu_{c^\star} - \mu_c)$, so for every $t$, we have:

$$0 \leq \mu_{c^\star} - \mu_{c_t} \leq \Delta_{\max}. \qquad \text{(ii)}$$

Therefore, we can write as:

$$\sum_{t \in \mathcal{T}_{\text{reg}}} (\mu_{c^\star} - \mu_{c_t}) \leq \Delta_{\max} |\mathcal{T}_{\text{reg}}|. \qquad \text{(iii)}$$

Since regulation rounds occur in $N$ disjoint episodes with lengths $\{\tau_{\text{reg}}^{(k)}\}_{k=1}^{N}$, it is equal to:

$$|\mathcal{T}_{\text{reg}}| = \sum_{k=1}^{N} \tau_{\text{reg}}^{(k)}. \qquad \text{(iv)}$$

Combining (iii) and (iv) gives us:

$$\sum_{t \in \mathcal{T}_{\text{reg}}} (\mu_{c^\star} - \mu_{c_t}) \leq \Delta_{\max} \sum_{k=1}^{N} \tau_{\text{reg}}^{(k)}. \qquad \text{(v)}$$

Now, on rounds $t \in \mathcal{T}_{\text{nor}}$, CORAL selects categories using the penalized score in Eq. (12). Define

$$Q_t(c) := \hat{r}_t(c) + \kappa\sqrt{\frac{\ln t}{\max(1, N_t(c))}} - \Lambda(D_t)\hat{R}_t^+(c). \quad \text{(vi)}$$

For brevity, let

$$b_t(c) := \kappa\sqrt{\frac{\ln t}{\max(1, N_t(c))}}. \qquad \text{(vii)}$$

Since $c_t$ maximizes $Q_t(c)$, for any oracle-safe comparator category $c^\star$, we have:

$$\begin{aligned} \hat{r}_t(c_t) &+ b_t(c_t) - \Lambda(D_t)\hat{R}_t^+(c_t) \\ &\geq \hat{r}_t(c^\star) + b_t(c^\star) - \Lambda(D_t)\hat{R}_t^+(c^\star). \end{aligned} \qquad \text{(viii)}$$

To bound the non-regulation regret, let's define the concentration event as:

$$\mathcal{E} := \{\forall t \leq T, \forall c \in \mathcal{C} : |\hat{r}_t(c) - \mu_c| \leq b_t(c)\}. \qquad \text{(ix)}$$

By Hoeffding's inequality and a union bound over $t \leq T$ and $c \in \mathcal{C}$, the event $\mathcal{E}$ holds with probability at least $1 - \delta$ for a suitable choice of $\kappa$. We now condition on $\mathcal{E}$.

Using (viii) and the fact that $\hat{R}_t^+(c_t) \geq 0$, we obtain:

$$
\begin{aligned}
\mu_{c^\star} &\leq \hat{r}_t(c^\star) + b_t(c^\star) \\
&\leq \hat{r}_t(c_t) + b_t(c_t) + \Lambda(D_t)\hat{R}_t^+(c^\star) \qquad \text{(x)} \\
&\leq \mu_{c_t} + 2b_t(c_t) + \Lambda(D_t)\hat{R}_t^+(c^\star).
\end{aligned}
$$

Therefore,

$$
\mu_{c^\star} - \mu_{c_t} \leq 2b_t(c_t) + \Lambda(D_t)\hat{R}_t^+(c^\star). \qquad \text{(xi)}
$$

Summing (xi) over all non-regulation rounds gives us:

$$
\sum_{t \in \mathcal{T}_{\text{nor}}} \left(\mu_{c^\star} - \mu_{c_t}\right) \leq 2 \sum_{t \in \mathcal{T}_{\text{nor}}} b_t(c_t) + \sum_{t \in \mathcal{T}_{\text{nor}}} \Lambda(D_t)\hat{R}_t^+(c^\star).
$$
$$\text{(xii)}$$

Define the cumulative proxy-risk penalty of the oracle-safe comparator as

$$
B_T(c^\star) := \sum_{t \in \mathcal{T}_{\text{nor}}} \Lambda(D_t)\hat{R}_t^+(c^\star). \qquad \text{(xiii)}
$$

It remains to bound the exploration term. Let $N_T(c)$ denote the number of times category $c$ is selected during non-regulation rounds. Since $\ln t \leq \ln T$, we have:

$$
\begin{aligned}
\sum_{t \in \mathcal{T}_{\text{nor}}} b_t(c_t) &\leq \kappa\sqrt{\ln T} \sum_{c \in \mathcal{C}} \sum_{s=1}^{N_T(c)} \frac{1}{\sqrt{s}} \\
&\leq 2\kappa\sqrt{\ln T} \sum_{c \in \mathcal{C}} \sqrt{N_T(c)} \qquad \text{(xiv)} \\
&\leq 2\kappa\sqrt{|\mathcal{C}|T\ln T}.
\end{aligned}
$$

Substituting (xiii) and (xiv) into (xii), we get:

$$
\sum_{t \in \mathcal{T}_{\text{nor}}} \left(\mu_{c^\star} - \mu_{c_t}\right) \leq O\left(\kappa\sqrt{|\mathcal{C}|T\ln T}\right) + B_T(c^\star). \quad \text{(xv)}
$$

Finally, combining the non-regulation regret bound in (xv) with the regulation-round bound in (v), we obtain:

$$
\mathfrak{R}(T) \leq O\left(\kappa\sqrt{|\mathcal{C}|T\ln T}\right) + B_T(c^\star) + \Delta_{\max} \sum_{k=1}^{N} \tau_{\text{reg}}^{(k)}.
$$
$$\text{(xvi)}$$

Hence Proved. $\qquad \square$

*Table 6.* Summary of Notations

| Notation | Description |
|---|---|
| **General Sets and Indices** | |
| $\mathcal{U}, \mathcal{I}, \mathcal{C}$ | Sets of users, items, and categories |
| $u, a, c$ | Indices for user, item, and category |
| $c(a)$ | Category to which item $a$ belongs |
| $\mathcal{I}_c$ | Set of items belonging to category $c$ |
| $T$ | Time horizon |
| $t$ | Discrete time step index |
| $\mathcal{H}_t$ | Interaction history up to time $t$, $\mathcal{H}_t = \{(a_1, y_1), \ldots, (a_t, y_t)\}$ |
| $\lambda_{u,c}(t)$ | Conditional intensity of user $u$ for category $c$ at time $t$ |
| $\mu_{u,c}$ | Base preference rate of user $u$ for category $c$ |
| $\beta_{u,c}$ | Resilience parameter (decay rate) for user $u$ and category $c$ |
| $\alpha_{c,c'}$ | Susceptibility coefficient (influence of category $c'$ on $c$) |
| $A_u$ | Susceptibility matrix for user $u$, $A_u = [\alpha_{c,c'}]_{|\mathcal{C}|\times|\mathcal{C}|}$ |
| $e_k$ | Engagement event indicator at step $k$ |
| $D_t$ | Exposure saturation state (cumulative load) at time $t$ |
| $\lambda_{\max}$ | Pre-defined exposure safety threshold |
| $L_t$ | Safety violation indicator, $L_t = \mathbb{I}(D_t > \lambda_{\max})$ |
| $R_t(c)$ | Conditional violation risk for category $c$ |
| $\hat{R}_t(c)$ | Empirical violation rate estimate |
| $\hat{R}_t^+(c)$ | Upper confidence bound on the violation risk |
| $\delta$ | Risk tolerance level (confidence parameter) |
| $N_t(c)$ | Number of times category $c$ has been selected up to time $t$ |
| $\Lambda(D_t)$ | State-dependent adaptive penalty function |
| $\gamma$ | Tolerance factor in saturation definition |
| $\kappa$ | Exploration constant |

*Table 7.* Statistics of the datasets.

| Dataset | # Users | # Items | # Interacts | Avg. Len | Density |
|---|---|---|---|---|---|
| Amazon | 65,262 | 106,427 | 1,540,690 | 23.61 | 0.02% |
| ML-1M | 6,040 | 3,416 | 999,611 | 165.50 | 4.84% |
| Steam | 39,795 | 10,587 | 1,790,393 | 44.99 | 0.42% |

*Table 8.* Ablation study across three datasets. We report SCC and CR. Bold indicates the best performance.

| METHOD | AMAZON | | ML-1M | | STEAM | |
|---|---|---|---|---|---|---|
| | COVERAGE ↑ | REWARD ↑ | COVERAGE ↑ | REWARD ↑ | COVERAGE ↑ | REWARD ↑ |
| CORAL (FULL) | **0.3443** | **0.2587** | **0.2086** | **0.2882** | **0.3015** | **0.5490** |
| W/O HAWKES | 0.3025 | 0.2184 | 0.1795 | 0.2453 | 0.2651 | 0.4776 |
| W/O RISK | 0.3218 | 0.2343 | 0.1942 | 0.2618 | 0.2834 | 0.5082 |
| W/O ADAPTIVE: | 0.3189 | 0.2301 | 0.1988 | 0.2694 | 0.2902 | 0.5195 |
| W/O EXPLORE | 0.2547 | 0.2491 | 0.1421 | 0.2620 | 0.2118 | 0.5114 |

