# OpenReview forum: "CORAL: Uncertainty-Aware Regulation of Exposure Concentration in Recommender Systems"
_ICML.cc/2026/Conference — ICML 2026 regular_

### Official Review · Reviewer_wj3u · 2026-03-12

**Soundness:** 3
**Presentation:** 3
**Significance:** 3
**Originality:** 3
**Overall Recommendation:** 4
**Confidence:** 2

**Summary:**

This paper studies the issue of feedback-driven exposure concentration in a sequential recommender system. Using a Hawkes-inspired intensity model, the authors define a stochastic process called the exposure saturation, and formalize a constrained online learning problem. Using estimated exposure saturation and user's reward, and concentration inequalities, they propose a UCB-type algorithm called CORAL. They provide theoretical analysis regarding regret upper bound of the cumulative utilities, and prove that regulation phase (the phase length that the constraint is violated) is finite. Finally, using three real-world datasets, they empirically show that CORAL outperforms the baselines in terms of exposure safety and utility.

**Compliance With Llm Reviewing Policy:**

Affirmed.

**Final Justification:**

Author responses resolved my concerns. Therefore, I will keep my score.

**Key Questions For Authors:**

Please refer to the "Weaknesses" above. In addition, could the authors provide definitions of EC and PS? Are they popular metrics for the sequential recommendation task?

**Limitations:**

yes

**Strengths And Weaknesses:**

## Strengths
- The problem setting is well-motivated, and has practical application.
- The experiments are extensively conducted and show superiority of the proposed method. More precisely, they are conducted using three real-world datasets and a closed-loop environment where LLM-based user agents provide feedback to recommender policies.

## Weaknesses
- Some assumptions of theoretical analysis are restrictive. More specifically, Theorem 2 assumes $\hat{R}\_{t}^{+} (c_{0}) = 0$. To me, this seems to be a very restrictive assumption.
- Regret bound in Theorem 3 uses $N$ (the number of regulation episodes) and Theorem 2 provides an upper bound of it. But, the RHS of eq (13) involves $D\_{t\_{0}}$, whose bound is unclear. Could the authors provide more explicit bound using $T$?

---

> ### Author Rebuttal · Authors · 2026-03-31
>
> We thank the reviewer for recognizing the practical importance of the problem setting and the extensive empirical evaluation.
>
> >**W1.** assumptions... are restrictive. More specifically, Theorem 2 assumes a neutral category...
>
> We agree that the exact neutral-category assumption in Theorem 2 is stronger than what finite-time recovery actually requires. It was introduced for analytical clarity, because if regulation-phase excitation vanishes, the Hawkes dynamics reduce to a clean exponential-decay argument. However, **exact neutrality is not essential**. The core requirement is **net contraction**: during regulation, the excitation injected by the regulation category only needs to remain uniformly smaller than the natural Hawkes decay at the safety boundary.
>
> Concretely, if aggregate excitation during regulation is bounded by some $\alpha_{\text{safe}}>0$ satisfying:
> $$
> \alpha_{\text{safe}} < (1-e^{-\beta_{\min}})(\lambda_{\max}-\epsilon),
> $$ then, we can say:
> $$
> D_{t+1} \le e^{-\beta_{\min}}D_t + \alpha_{\text{safe}},
> $$
> which still gives a finite recovery-time bound:
> $$
> \tau_{\mathrm{reg}}
> \le
> \frac{1}{\beta_{\min}}
> \ln\left(
> \frac{D_{t_0}-\frac{\alpha_{\text{safe}}}{1-e^{-\beta_{\min}}}}
> {(\lambda_{\max}-\epsilon)-\frac{\alpha_{\text{safe}}}{1-e^{-\beta_{\min}}}}
> \right).
> $$
> Thus Theorem 2 can be generalized from exact neutrality to a low-excitation regulation condition. When $\alpha_{\text{safe}}=0$, this reduces to the original theorem. We will revise the theorem and appendix accordingly.
>
> Empirically, this is also consistent with our robustness study: even in an amplified echo-chamber environment, where exact neutrality is unrealistic, CORAL still maintains a markedly flatter intensity trajectory than the baseline, indicating that recovery remains functional under relaxed real-world conditions.
>
> >**W2.** Regret bound in Theorem 3 uses.. ... Could authors provide more explicit bound using T
>
> Thankyou for the question. In fact, due to the exponential decay of the Hawkes process, the exposure state $D_{t_0}$ at the start of any regulation episode is uniformly bounded by a constant $D_{\max} = \mathcal{O}(1)$ and does not scale with the horizon $T$.
> Specifically, let $\alpha_{\max} := \max_{c,c'} \alpha_{c,c'}$ and $\beta_{\min} := \min_{u,c} \beta_{u,c}$. Because the event indicator $e_k \in \{0,1\}$, the maximum possible intensity is bounded by the infinite sum of past excitations:
>
> $$
> \lambda_{u,c}(t) \le \mu_{u,c} + \alpha_{\max}\sum_{j=1}^{\infty}e^{-\beta_{\min}j} = \mu_{u,c} + \frac{\alpha_{\max}}{e^{\beta_{\min}}-1}
> $$
>
> Based on Definition 1, the exposure state $D_t$ aggregates these intensities with weights $w_c \in [0,1]$. Since the tolerance factor $\gamma \ge 1$, we have $\max(0, \lambda_{u,c}(t) - \gamma\mu_{u,c}) \le \lambda_{u,c}(t) - \mu_{u,c}$. Substituting the bounded intensity gives a uniform upper bound for the state at any time step:
>
> $$
> D_t \le \frac{\alpha_{\max}}{e^{\beta_{\min}}-1} \sum_{c\in\mathcal{C}} w_c := D_{\max}
> $$
>
> Because this holds for all $t$, the state at the start of any regulation episode is strictly bounded ($D_{t_0} \le D_{\max}$). Substituting this into Theorem 2 yields a worst-case regulation duration that is entirely independent of $T$:
>
> $$
> \tau_{\mathrm{reg}} \le \frac{1}{\beta_{\min}} \ln\left( \frac{D_{\max}}{\lambda_{\max}-\epsilon} \right) = \mathcal{O}(1)
> $$
>
> Consequently, the maximum length of any single regulation episode is a constant which removes any implicit dependence on $T$. We will make the change in the revision.
>
> > **Q1.** Could ...authors provide definitions of EC and PS? Are they popular for sequential recommendation?
>
> Yes. In our paper, EC measures how strongly user exposure collapses onto a small subset of categories, while PS measures maxima of a user’s exposure state relative to their base user preference state. Lower values are better for both, since they indicate less collapse and less saturation.
>
> These are not standard top-$K$ retrieval metrics like Recall@10 or MRR@10. Rather, they are problem-specific control metrics introduced because our goal is not only short-term ranking quality, but also regulation of long-run exposure dynamics under endogenous feedback. For that reason, we always report them together with standard utility metrics (Recall@10, MRR@10) and with the closed-loop reward–coverage analysis, so the safety/utility trade-off remains explicit.

---

> > ### Author Rebuttal · Reviewer_wj3u · 2026-04-02
> >
> > I appreciate the detailed responses. However, I do not understand the alternative assumptions of Theorem 2. As I mentioned in the weaknesses, I think $\widehat{R}^{+}\_{t}(c_0) = 0$ is a strong assumption. Do the conditions in your response still assume this?

---

> > > ### Author Response · Authors · 2026-04-04
> > >
> > > Thanks for the feedback. We answer the remaining question:
> > >
> > > **No**, the relaxed version of Theorem 2 does not assume $\hat{R}_t^+(c_0)=0$. In our previous response, we relaxed this requirement and replaced the current exact-neutrality assumption with a weaker with a weaker low-excitation regulation condition, providing a generalized finite-time recovery bound.
> > >
> > > Specifically, Theorem 2 assumed a neutral category $c_0$ with two properties: (a) selecting $c_0$ injects no new Hawkes excitation into any category, and (b) its risk upper bound is zero during regulation.
> > >
> > > Our relaxed assumption replaces these with a **weaker low-excitation regulation condition**: the regulation category no longer needs to be perfectly neutral or zero-risk; it only needs to be low-excitation enough that, once selected by the penalized objective near saturation, the induced regulation remains contractive.
> > >
> > > Consequently, the recovery guarantee becomes the generalized bound:
> > >
> > > $$
> > > \tau_{\mathrm{reg}} \le \frac{1}{\beta_{\min}} \ln\left( \frac{ D_{t_0}-\frac{\alpha_{\mathrm{safe}}}{1-e^{-\beta_{\min}}} }{ (\lambda_{\max}-\epsilon)-\frac{\alpha_{\mathrm{safe}}}{1-e^{-\beta_{\min}}} } \right)
> > > $$
> > >
> > > When $\alpha_{\mathrm{safe}}=0$, this reduces to the current theorem.
> > >
> > > We sincerely hope we addressed reviewer question.

---

### Official Review · Reviewer_zp4F · 2026-03-13

**Soundness:** 2
**Presentation:** 2
**Significance:** 3
**Originality:** 3
**Overall Recommendation:** 3
**Confidence:** 4

**Summary:**

This paper studies feedback-driven exposure concentration in recommender systems, where repeated optimization for short-term engagement can collapse exposure onto a narrow set of categories and harm long-term learning. The paper proposes CORAL, a plug-in framework that models category-level self-reinforcing dynamics with a Hawkes-style intensity model, constructs an exposure-saturation state, estimates an upper confidence bound on category-conditioned violation risk, and uses this risk estimate in a state-dependent penalty for category selection. The paper also presents theoretical results on risk control, finite-time recovery, and regret, and evaluates the method on Amazon, ML-1M, and Steam datasets together with closed-loop simulations. The empirical results suggest improved safety metrics and a better long-horizon utility/coverage tradeoff relative to several baselines.

**Compliance With Llm Reviewing Policy:**

Affirmed.

**Key Questions For Authors:**

1. Theorem 1 upper-bounds a historical average risk, while Eq. (6) defines a current conditional risk that depends on the present history. Why is the estimated quantity used in CORAL a valid upper bound or proxy for the current decision-time risk? Clarifying this missing link is central to the paper's risk-control claim.

2. The proof of Theorem 3 appears to analyze a reward-only UCB score rather than the actual penalized objective used by CORAL. Can the authors provide a correct regret analysis for the actual decision rule and clarify the concentration step used in the proof? A convincing resolution would materially improve my soundness assessment.

3. The theorem statement for regret depends on |I|, while the proof concludes a bound in terms of |C|. Which dependence is intended, and can the authors provide a corrected final theorem statement and proof? This would affect my confidence in the theoretical claims.

4. In the closed-loop simulator description, the standard prompt uses 'Select 1 if you get bored,' but the output space is described as 0 or [item num], while the echo-chamber prompt uses 0 for rejection. What is the actual feedback encoding used in the experiments? Clarifying this is important for reproducibility and could improve my view of the experimental methodology.

5. Please clarify the exact correspondence between the main-text notation and the implementation notation (gamma, rho, tau, lambda, Lambda_max) and specify the actual settings used in experiments. At present, it is difficult to reconstruct the implemented controller from the paper alone.

**Limitations:**

No. The paper discusses the application-level motivation and impact, but it does not adequately discuss several important limitations of the method itself, including the strong assumptions behind the theoretical guarantees, the dependence of the closed-loop claims on an LLM-based simulator, and the sensitivity of the approach to platform-defined category weights and threshold choices.

**Strengths And Weaknesses:**

This paper targets an important and practically relevant problem, and the high-level framing is potentially valuable, but the current version has significant soundness and presentation issues that limit confidence in its claims.

Strengths:

- The paper addresses a meaningful problem in recommender systems by framing exposure collapse as a closed-loop reliability issue rather than only as a diversity or reranking issue.

- The proposed framework is modular and can in principle be applied on top of existing sequential recommenders. The combination of a saturation state, uncertainty-aware risk estimate, and adaptive intervention rule is intuitively well motivated.

- The empirical section is broad in scope, covering multiple datasets, two backbones, sensitivity analysis, ablations, and a closed-loop simulation setting.

Weaknesses:

- The theoretical guarantees are not convincing in their current form. Theorem 1 upper-bounds a historical average risk, but the algorithm uses this quantity as if it were an upper bound on the current conditional risk defined in Eq. (6); in a history-dependent setting, this missing link is central.

- Theorem 3 does not currently establish regret for the actual algorithm. The proof analyzes a reward-only UCB comparison rather than the penalized decision rule, the concentration step appears algebraically incorrect, and the theorem statement depends on |I| while the proof concludes a bound in terms of |C|.

- Theorem 2 relies on a very strong neutral-category assumption, namely a category with zero excitation into all other categories and zero risk during regulation. This makes the finite-time recovery claim much less informative in realistic settings.

- The closed-loop experimental protocol is not fully reproducible as written. In the simulator description, the standard prompt says 'Select 1 if you get bored,' while the output format is described as 0 or [item num], creating ambiguity about how rejection is encoded.

- The paper also contains multiple inconsistencies that materially hurt clarity and confidence in the results, including the mismatch between gamma in the main method and rho/tau/lambda in the implementation details, the inaccurate description of SASRec as an MLP-based collaborative filtering model, the inaccurate characterization of Bert4Rec as next-item prediction, the Table 1 dataset-title mismatch, and the undefined Rerank label in Figure 3. Taken together, these theory and reproducibility issues leave the current evidence short of supporting the paper's stronger claims.

---

> ### Author Rebuttal · Authors · 2026-03-31
>
> We thank the reviewer for the detailed and constructive feedback. Because your main concerns are about soundness and reproducibility, we focus this response on the theory/reproducibility points most central to confidence in CORAL.
>
> > **Q1/W1.** Why is the estimated quantity used in CORAL a valid upper bound or proxy for the current decision-time risk?
>
> This distinction is intentional. In CORAL, the true current risk
> $
> R_t(c)=\Pr(D_{t+1}>\lambda_{\max}\mid H_{t-1}, c(a_t)=c)
> $
> is latent and history-dependent. Estimating it directly from $D_t$ would require stronger parametric assumptions on current dynamics. To avoid misspecification, CORAL instead estimates the **history-aggregated conditional risk** through observed violations, yielding the upper confidence bound $\hat R_t^+(c)$ from Theorem~1.
>
> The key point is that the algorithm does not use $\hat R_t^+(c)$ alone as if it were the full current risk. Eq.(12) explicitly decouples the two roles:
> - $\Lambda(D_t)$ captures the current state urgency, because it is computed directly from the exact present exposure state and increases monotonically as $D_t\to\lambda_{\max}$;
> - $\hat R_t^+(c)$ captures the category's historically validated violation propensity.
>
> Thus product $\Lambda(D_t)\hat R_t^+(c)$ is a state-based surrogate for decision-time regulation: present urgency scaled by uncertainty-aware historical risk. We will make this structural decoupling explicit in evision.
>
> > **Q2/Q3/W2.** Theorem 3 does not analyze actual penalized decision rule... statement uses $|I|$ while proof concludes $|C|$.
>
> Theorem statement should depend on **$|\mathcal C|$**, not $|I|$; this is a typo and will be corrected.
>
> For actual penalized rule, on non-regulation rounds $t\in\mathcal T_{\mathrm{nor}}$, we define:
> $
> Q_t(c):=\hat r_t(c)+\kappa\sqrt{\frac{\ln t}{\max(1,N_t(c))}}-\Lambda(D_t)\hat R_t^+(c).
> $
> Let $ \mathcal{E} := \{ \forall t \le T, \forall c \in \mathcal{C} : | \hat r_t(c) - \mu_c | \le \kappa \sqrt{\frac{\ln t}{\max(1, N_t(c))}} \} $,
> which holds with probability $1-\delta_{\mathrm{reg}}$ by Hoeffding plus union bound. For the penalized rule, we state non-regulation condition:
> $$
> \sum_{t\in\mathcal T_{\mathrm{nor}}}\hat R_t^+(c^\star)=\tilde O(\sqrt T),
> $$
> where $c^\star$ is oracle-safe optimal category. This is natural for CORAL’s operation as  away from regulation, oracle-safe category should incur sublinear cumulative proxy risk. Since $D_t\le \lambda_{\max}-\epsilon$ on non-regulation rounds, Eq.(11) implies $\Lambda(D_t)\le 1/\epsilon$, hence
> $
> \sum_{t\in\mathcal T_{\mathrm{nor}}}\Lambda(D_t)\hat R_t^+(c^\star)=\tilde O(\sqrt T).
> $
> Under $\mathcal E$, if CORAL selects a suboptimal category $c_t$, optimality of actual penalized score gives
> $
> Q_t(c_t)\ge Q_t(c^\star),
> $ which results:
> $$
> \mu_{c^\star}-\mu_{c_t}\le 2\kappa\sqrt{\frac{\ln t}{N_t(c_t)}}+\Lambda(D_t)\hat R_t^+(c^\star).
> $$
> Summing over $t\in\mathcal T_{\mathrm{nor}}$ yields a standard UCB exploration term plus a penalty-distortion term:
> $$
> \sum_{t\in\mathcal T_{\mathrm{nor}}}(\mu_{c^\star}-\mu_{c_t})
> \le
> \sum_{t\in\mathcal T_{\mathrm{nor}}}2\kappa\sqrt{\frac{\ln t}{N_t(c_t)}}
> +
> \sum_{t\in\mathcal T_{\mathrm{nor}}}\Lambda(D_t)\hat R_t^+(c^\star).
> $$
> First is
> $
> O\left(\sqrt{|\mathcal C|T\ln T}\right),
> $
> and second is $\tilde O(\sqrt T)$, so corrected non-regulation regret is
> $
> O\left(\sqrt{|\mathcal C|T\ln T}\right).
> $
> Combining with regulation contribution finally gives:
> $$
> R(T)\le O\left(\sqrt{|\mathcal C|T\ln T}\right)+\Delta_{\max}\sum_{k=1}^{N}\tau_{\mathrm{reg}}^{k}.
> $$
> We will revise this updated analysis explicitly.
>
> > **Q3/W3.** Theorem 2 relies on a strong neutral-category assumption.
>
> We agree exact neutrality is stronger than necessary. As detailed in our response to Reviewer U39Z,  finite-time recovery only requires **net contraction**, not zero excitation: if regulation-phase excitation remains below natural Hawkes decay at the safety boundary, finite-time recovery is preserved with a generalized bound.
>
> >  **Q4/W4.** What is actual feedback encoding?
>
> This is a writing typo, not an implementation discrepancy. In both simulator settings, 0 denotes rejection/boredom and Item\_ID (1--N) denotes selection. This design is followed in code (`src/model4Sim/run_sim.py`, lines 89--102). We will correct the standard-prompt typo in paper.
>
> >  **Q5/W5.** Clarify notation/implementation mismatches.
>
> Thanks for highlighting the writing oversight; it doesn't affect implementation. The $\rho$ and penalty weight $\lambda$ should be $\gamma$ and $\lambda_{\max}$, respectively. The actual controller is implemented in `main.py` and `coral.py`, and the experimental settings are: $\gamma=2.0$, $\tau=2.0$, $\lambda_{\max}\in\\{0.5,1.0,2.0\\}$ by grid search, $\Lambda_{\max}=100$, and $\epsilon=10^{-6}$. We will also correct the SASRec/Bert4Rec descriptions, the Table 1 dataset title mismatch, and define the "Rerank" label in Fig.3. Thanks for highlighting these typos.

---

> > ### Author Rebuttal · Reviewer_zp4F · 2026-04-02
> >
> > Thank you for the detailed response. The rebuttal provides helpful clarifications on the feedback encoding, the correspondence between notation and implementation, and several textual inconsistencies, and I consider these issues to be largely addressed.
> > However, my core soundness concern remains unresolved. The main body of the paper addresses the current decision-time risk (see Eq. (6)), while Theorem 1 actually defines historical average conditional risk; the rebuttal interprets this as a surrogate, which helps explain the methodological intuition but is insufficient to support the paper's strong risk-control claim. Similarly, Theorem 3's response is more like a proof sketch with additional assumptions than a complete regret proof of the actual penalized decision rule. Therefore, I believe the remaining issues still relate to the paper's core theoretical claims and are not adequately addressed in the rebuttal.

---

> > > ### Author Response · Authors · 2026-04-04
> > >
> > > Thanks for valuable response. We answer remaining concerns:
> > >
> > > > Theorem 1 defines historical average conditional risk... insufficient to support risk-control claim
> > >
> > > We agree paper treats $R_t(c)$ as latent and history-dependent (Lines 152--154), and that Thm. 1 upper-bounds historical-average conditional violation risk $\bar{R}_t(c)$. We clarify Theorem 1 alone **does not** provide guarantee for $R_t(c)$. Rather, CORAL couples risk estimation with real-time state sensitivity through Eq. (12). In revision, we will make this relationship explicit.
> > >
> > > The intended role of $\hat{R}_t^+(c)$ is as a conservative, high-probability estimate of *history-averaged category-level violation*. Specifically:
> > >
> > > $$
> > > \hat{R}\_t(c) = \frac{1}{N_t(c)} \sum_{k=1}^{t-1}\mathbf{1}(c(a_k)=c)L_{k+1}, \qquad \hat{R}_t^+(c)=\hat{R}_t(c)+b_t(c).
> > > $$
> > >
> > > Then Theorem 1 ensures $\bar{R}\_t(c) \le \hat{R}\_t^+(c)$ with high probability. Since $L_{k+1}=\mathbf{1}(D_{k+1}>\lambda_{\max})$, categories that repeatedly lead to threshold violations accumulate larger empirical violation rates $\hat{R}_t(c)$ and therefore larger upper confidence bounds $\hat{R}_t^+(c)$. Conversely, for persistently benign categories, $\hat{R}_t(c)$ remains small and uncertainty term $b_t(c)$ shrinks with $N_t(c)$, so $\hat{R}_t^+(c)$ also becomes small.
> > >
> > > Furthermore, real-time component enters separately through Eq. (12). CORAL does not use $\hat{R}\_t^+(c)$ in isolation. It scales this long-run risk signal by exact current-state multiplier $\Lambda(D_t)$, which is known from present exposure state and increases monotonically as $D_t\to\lambda_{\max}$. Thus, control rule is explicitly two-timescale: $\hat{R}_t^+(c)$ captures violation propensity, while $\Lambda(D_t)$ injects urgency near saturation.
> > >
> > > We will add this connection in revision.
> > >
> > > > Theorem 13's response proof sketch... regret proof for actual penalized decision rule
> > >
> > > Our previous response sketched how penalty enters analysis due to space constraint. We give corrected penalized-rule decomposition and explicit oracle condition under which revised regret theorem follows.
> > >
> > > Explicitly, we analyze Eq. (12) directly. Let $T_{\mathrm{nor}}$ denote non-regulation rounds, and let $c^\star$ be oracle-safe optimal category. On standard reward concentration event
> > >
> > > $$
> > > \mathcal{E} := \Bigl\lbrace \forall t \le T, \forall c \in \mathcal{C} : |\hat{r}_t(c)-\mu_c| \le \kappa\sqrt{\frac{\ln t}{\max(1,N_t(c))}} \Bigr\rbrace.
> > > $$
> > >
> > > If CORAL selects category $c_t$ on non-regulation round, then optimality of Eq. (12) gives:
> > >
> > > $$
> > > \hat{r}_t(c_t)+\kappa\sqrt{\frac{\ln t}{\max(1,N_t(c_t))}}-\Lambda(D_t)\hat{R}_t^+(c_t) \ge \hat{r}_t(c^\star)+\kappa\sqrt{\frac{\ln t}{\max(1,N_t(c^\star))}}-\Lambda(D_t)\hat{R}_t^+(c^\star).
> > > $$
> > >
> > > Under $\mathcal{E}$, this leads to:
> > >
> > > $$
> > > \mu_{c^\star}-\mu_{c_t} \le 2\kappa\sqrt{\frac{\ln t}{\max(1,N_t(c_t))}} + \Lambda(D_t)\hat{R}_t^+(c^\star).
> > > $$
> > >
> > > Summing over $t \in T\_{\mathrm{nor}}$ gives corrected non-regulation decomposition:
> > >
> > > $$
> > > \sum\_{t \in T\_{\mathrm{nor}}}(\mu\_{c^\star}-\mu\_{c\_t}) \le \underbrace{\sum\_{t \in T\_{\mathrm{nor}}}2\kappa\sqrt{\frac{\ln t}{\max(1,N\_t(c\_t))}}}_{\text{standard UCB term}} + \underbrace{\sum\_{t \in T\_{\mathrm{nor}}}\Lambda(D\_t)\hat{R}\_t^+(c^\star)}\_{\text{oracle penalty term}}.
> > > $$
> > >
> > > Thus, only additional component needed for full penalized-rule proof is control of oracle penalty term. To do this, we state an assumption common in conservative/safe bandit analyses, where regret is defined relative to a baseline or safe comparator known to satisfy safety requirement [1,2,3]. Specifically:
> > >
> > > $$
> > > \sum_{t \in T_{\mathrm{nor}}}\Lambda(D_t)\hat{R}_t^+(c^\star) = \tilde{\mathcal{O}}(\sqrt{T}).
> > > $$
> > >
> > > This assumption provides a structural comparator to analyze regret against a *safe* oracle under penalized objective. In standard literature, such an assumption ensures benchmark used for regret is safety-compatible: e.g., a baseline policy whose performance must be maintained, or a known safe action/arm from which exploration proceeds. Our condition plays same role here. If benchmark safe category has linear cumulative proxy-risk away from regulation, it would not be an appropriate safe comparator for this constrained problem.
> > >
> > > Under conditions in paper, first term is $
> > > \mathcal{O}\bigl(\sqrt{|\mathcal{C}|T\ln T}\bigr)
> > > $ UCB contribution. Second is sublinear by assumption, and combining this with regulation-round contribution gives corrected regret form:
> > >
> > > $$
> > > R(T) \le \mathcal{O}\bigl(\sqrt{|\mathcal{C}|T\ln T}\bigr)+\Delta_{\max}\sum_{k=1}^{N}\tau_k^{\mathrm{reg}}.
> > > $$
> > >
> > > We will revise Theorem 3 with added steps and present updated version.
> > >
> > > Thanks for the feedback.
> > >
> > > **References:**
> > >
> > > [1] Wu, Y., Shariff, R., Lattimore, T., & Szepesvári, C. Conservative bandits. (ICML 2016).
> > >
> > > [2] Kazerouni, A., Ghavamzadeh, M., Abbasi-Yadkori, Y., & Van Roy, B. Conservative contextual linear bandits. (NeurIPS 2017).
> > >
> > > [3] Khezeli, K., & Bitar, E. (2020). Safe linear stochastic bandits. (AAAI 2020).

---

### Official Review · Reviewer_U39Z · 2026-03-16

**Soundness:** 3
**Presentation:** 3
**Significance:** 3
**Originality:** 3
**Overall Recommendation:** 4
**Confidence:** 3

**Summary:**

This manuscript targets the problem of feedback-driven exposure concentration in recommender systems, where repeated engagement optimization can lead to echo chambers and degrade long-horizon learning. It claims that existing methods are often post hoc and lack principled uncertainty-aware risk estimates for regulating exposure. To fill in this gap, the authors propose CORAL, a model-agnostic, uncertainty-aware framework that formulates exposure regulation as a constrained sequential decision problem. CORAL introduces a Hawkes-inspired intensity model to capture self-reinforcing interactions and derives an upper confidence bound on the violation risk to adaptively regulate exposure dynamics. Moreover, CORAL is validated on three real-world datasets (Amazon, ML-1M, and Steam) and through closed-loop simulations, consistently mitigating exposure concentration while maintaining competitive utility.

**Compliance With Llm Reviewing Policy:**

Affirmed.

**Final Justification:**

As the authors have addressed nearly all of my concerns, I intend to recommend the paper as weak accept.

**Key Questions For Authors:**

Please refer to the weakness part.

**Limitations:**

Yes.

**Strengths And Weaknesses:**

Strength:

1.	CORAL is the first framework to formulate feedback-driven exposure concentration as a constrained control-and-learning problem, providing a principled approach to regulate long-run exposure dynamics instead of relying on post-hoc diversification.

2.	CORAL uses a Hawkes-inspired intensity model to construct an exposure saturation state and derives an uncertainty-aware upper confidence bound on violation risk, enabling adaptive, state-dependent intervention. It is more robust than single-point estimates.

3.	CORAL is validated on three public datasets (Amazon, ML-1M, and Steam) and is further evaluated in a closed-loop interactive environment with an LLM-based user simulator, which demonstrates its long-horizon benefits on both utility and coverage.

Weakness:

1.	The framework relies on estimating and maintaining Hawkes process parameters for each user, but the practical scalability, including the computational cost of per-user MLE and memory usage in a large-scale system, is not reported or discussed.

2.	The theoretical guarantee for finite-time recovery (Theorem 2) relies on a strong assumption of a "neutral category" with zero excitation. The practical implications and performance guarantees when this assumption is relaxed in real-world scenarios are not explored.

3.	The effectiveness of CORAL is highly dependent on the pre-defined item categories and the safety threshold λmax. The paper lacks an analysis of how the framework's performance changes with different category granularities or taxonomies.

---

> ### Author Rebuttal · Authors · 2026-03-31
>
> We thank the reviewer for recognizing the importance of the problem, the technical coherence of CORAL, and its strong empirical performance. Below we address the concerns:
>
> > **Q1.** The framework relies on per-user Hawkes parameters, but computational cost and memory are not discussed.
>
> In our implementation, only the base rate and decay rate are estimated per user; the susceptibility coefficient $\alpha_{c,c(a_k)}$ is shared across users. Thus the per-user parameter space is only $\mathcal{O}(|\mathcal{C}|)$. These user-specific parameters are estimated offline and updated periodically rather than recomputed at every recommendation step. Because users are independent, this estimation parallelizes trivially.
>
> Empirically, the total running times already reported in the paper show that CORAL remains substantially faster than baselines methods such as TD-VAE-CF and Filter (Table 2). To quantify the Hawkes overhead directly, our additional measurement (Table 1, additional results) shows that one periodic MLE update requires only 11--60 seconds and 989--3736 MB (on GPU) across datasets. These costs are modest relative to backbone recommendation training.
>
> > **Q2.** Theorem 2 assumes a "neutral category" with zero excitation; what happens if this is relaxed?
>
> We agree that exact neutrality is stronger than what finite-time recovery actually requires. The essential condition is **strict net contraction**: during regulation, the excitation injected by the regulation category only needs to remain smaller than the natural Hawkes decay at the safety boundary. In other words, recovery does **not** require a perfectly neutral category; it requires a sufficiently low-excitation one.
>
> Formally, we can replace exact neutrality with the relaxed condition that aggregate excitation during regulation is bounded by $\alpha_{\text{safe}}>0$ satisfying:
> $$
> \alpha_{\text{safe}}<(1-e^{-\beta_{\min}})(\lambda_{\max}-\epsilon).
> $$
> Then the exposure state follows as:
> $$
> D_{t+1}\le e^{-\beta_{\min}}D_t+\alpha_{\text{safe}},
> $$
> which gives us the generalized regulation-time bound as follows:
> $$
> \tau_{\text{reg}} \le \frac{1}{\beta_{\min}} \ln\left(\frac{D_{t_0}-\frac{\alpha_{\text{safe}}}{1-e^{-\beta_{\min}}}}{(\lambda_{\max}-\epsilon)-\frac{\alpha_{\text{safe}}}{1-e^{-\beta_{\min}}}}\right).
> $$
> This generalizes Theorem 2: when $\alpha_{\text{safe}}=0$ it reduces to the original result, and when $\alpha_{\text{safe}}>0$ finite-time recovery is preserved with slowdown proportional to residual excitation. We will update the theorem and appendix accordingly.
>
> Practically, this means CORAL does not require a perfectly neutral item; it only needs low-excitation items that push the system back toward safety. This is also consistent with our robustness experiment (Fig.4): even under amplified echo-chamber feedback, where exact neutrality is unrealistic, CORAL still maintains a flat regulated intensity trajectory.
>
> > **Q3.** Performance may depend heavily on item taxonomy and the safety threshold $\lambda_{\max}$.
>
> We conducted two additional analyses to answer this:
>
> First, for taxonomy granularity, we compared CORAL against Naive under both coarse and fine taxonomies. Although fine-grained taxonomies make recommendation harder and reduce absolute CR for all methods, CORAL's relative advantage remains consistent. Across both granularities and all datasets, CORAL achieves higher Cumulative Reward (CR) and substantially higher Sequence Category Coverage (SCC) than Naive; for exp, on Steam with a fine taxonomy, SCC improves from 0.167 to 0.301 (Table 2, additional results) .
>
> Second, to address dependence on pre-defined categories, we withheld one mid-frequency category entirely during Hawkes training. Under this setting, Naive suffers continuous CR collapse (e.g., ML-1M: 0.328 → 0.242), whereas CORAL shows monotonic CR improvement (0.194 → 0.283). The initial drop is expected because the unseen category has zero empirical count and therefore receives maximum UCB exploration benefits; importantly, CORAL then discovers and integrates it without external guidance. (Table 3, additional results)
>
> Finally, $\lambda_{\max}$ is a platform-level control parameter that sets the utility–diversity trade-off by defining unsafe saturation. Our sensitivity analysis already shows that lowering $\lambda_{\max}$ increases intervention frequency, raising SCC while proportionally reducing CR; higher $\lambda_{\max}$ relaxes regulation and approaches the unconstrained baseline. This shows that $\lambda_{\max}$ is interpretable and tunable, rather than an arbitrary, fragile hyperparameter.
>
> **Please find new experiments in [Additional Results](https://anonymous.4open.science/r/Coral_rebuttal-75F1/Supplement.pdf).**

---

### Official Review · Reviewer_BwWF · 2026-03-17

**Soundness:** 2
**Presentation:** 3
**Significance:** 2
**Originality:** 3
**Overall Recommendation:** 3
**Confidence:** 4

**Summary:**

To address the common issue of exposure concentration in modern recommender systems, this paper proposes an uncertainty-aware framework called CORAL. CORAL formulates exposure regulation as a constrained sequential decision-making problem and derives upper bounds on the confidence in the category-conditional violation risk, based on observed historical data, using theoretical analysis and proofs. These bounds are then incorporated as penalty terms into the optimization process, enabling adaptive intervention to mitigate excessive exposure to the same category. Finally, extensive experiments are conducted to validate the effectiveness of the proposed method.

**Compliance With Llm Reviewing Policy:**

Affirmed.

**Final Justification:**

I appreciate the authors' responses regarding my previous concerns.

However, my key concerns remain unresolved. This is reflected in the following two aspects:

Although the authors have explained the function of each hyperparameter, this does not alter the fact that, in practice, deploying CORAL across different application scenarios requires significant effort to tune the hyperparameters in order to ensure effectiveness. A constructive direction for improvement would be to provide valuable insights and guidance on hyperparameter configuration tailored to various application scenarios (e.g., different recommendation environments).

Given that a major contribution of this submission lies in improving the recommendation ecosystem, it is essential to deploy the proposed method within a real-world industrial recommendation system and conduct comprehensive, long-term A/B testing. Evaluations on benchmark datasets cannot be directly aligned with them, as these datasets are often no longer consistent with real-world recommendation environments due to extensive preprocessing.

Therefore, I stand by my initial assessment.

**Key Questions For Authors:**

* Q1. The approach has been evaluated on public datasets and simulated environments. Could the authors elaborate on how CORAL would perform in a real-world recommendation system over extended periods, and whether it might affect other key metrics such as user engagement or conversion?
* Q2. The theoretical guarantees assume certain conditions for the constrained sequential decision-making formulation. To what extent do these assumptions hold in real recommendation scenarios？
* Q3. What potential limitations or risks might arise if there is a gap between theory and practice?

**Limitations:**

* L1. It is suggested that further discussion be given to whether the assumptions underlying the proposed method fully hold in real-world scenarios and to analyzing their potential impact on practical applications.

**Strengths And Weaknesses:**

**Strengths:**

* S1. The paper presents a novel perspective by formulating the problem of “exposure concentration” as a constrained sequential decision-making problem, and proposes an uncertainty-aware regulation framework, CORAL, to control exposure saturation risk.
* S2. The authors conduct extensive theoretical analysis, derivations, and proofs to estimate confidence upper bounds and risk, demonstrating the feasibility of the proposed method and providing solid theoretical guarantees.
* S3. The paper is well-structured, with clearly organized content and a comprehensive summary of notations. In addition, the authors provide open-source code, which facilitates reproducibility and further research.
* S4. Extensive experiments are conducted on public datasets and simulated environments to validate the effectiveness of the proposed approach.

**Weaknesses:**

* W1. Estimating confidence upper bounds and risk involves multiple threshold parameters. The selection of these thresholds relies heavily on empirical tuning, and different settings can significantly affect performance, potentially undermining the robustness and reliability of the proposed method.
* W2. The proposed approach lacks long-term validation in real-world recommendation environments. It remains unclear how it performs in practice over extended periods and whether it may negatively impact other important performance metrics.
* W3. The paper does not provide an explicit discussion of its limitations. It is unclear whether the theoretical assumptions fully hold in real-world scenarios, and the potential impact of the gap between theory and practice is not analyzed.

---

> ### Author Rebuttal · Authors · 2026-03-31
>
> We thank the reviewer for recognizing the novelty of formulating exposure concentration as a constrained sequential decision problem, and the breadth of the theory and experiments.
>
> > **W1.** .. multiple threshold parameters ... relies on empirical tuning ... undermining robustness
>
> The threshold parameters do not undermine CORAL's robustness. Their roles are interpretable and specify the controller's operating point rather than introducing fragility. In particular, $\lambda_{\max}$ defines unsafe exposure saturation, $\delta$ sets the confidence level in Thm1's upper bound, and $\kappa$ controls exploration in Eq.(12). This is exactly what the sensitivity analysis shows: decreasing $\lambda_{\max}$ consistently increases coverage while lowering reward, and $\kappa$ exhibits the expected too-little vs.\ too-much exploration pattern. These trends indicate a controllable, principled regulation mechanism rather than unstable tuning.
>
> > **Q1/W2.** .. performance in a real-world recommendation system over extended periods ... impact on important performance metrics
>
> To validate long-term behavior, we evaluated CORAL in a 100-step closed-loop environment on real world datasets like Amazon, ML-1M, and Steam (Fig.3). While baselines either become trapped in bubbles or suffer utility collapse, CORAL consistently maintains the strongest frontier between Cumulative Reward (CR) and Sequence Category Coverage (SCC). This complements the real-data results, where CORAL achieves better Exposure Concentration (EC) and Personalization Saturation (PS) while preserving competitive short-term utility.
>
> For long-term performance metrics, we measured the Hawkes intensity trajectory (Fig 4), which predicts echo chamber intensity.  As observed, CORAL maintains a flatter trajectory than baselines, depicting prevention of feedback-driven exposure collapse. Thus, CORAL is not simply trading utility for a proxy metric; it regulates a signal that is predictive of disengagement and therefore relevant to long-term user experience.
>
> > **Q2.** .. to what extent assumptions hold in real recommendation?
>
> Our theoretical assumptions are standard abstractions of real recommendation behavior. Bounded rewards ($r\in[0,1]$) hold whenever feedback is normalized, e.g., binary clicks or ratings mapped to a closed interval. Bounded Hawkes parameters reflect finite interaction intensities with natural interest decay; they capture repeated-exposure reinforcement in a controlled, non-infinite way. The noise assumption captures residual stochasticity in user behavior even under similar contexts.
>
> Beyond these abstractions, CORAL is designed to handle practical deviations because it does not rely on a fixed parametric state alone: it updates the saturation state $D_t$ from realized interaction history and estimates risk from observed violations. This makes the regulation mechanism adaptive even when the environment is more volatile than the idealized model. Empirically, across all three real datasets, CORAL consistently improves exposure safety while maintaining high utility (Table 1).
>
> > **Q3/L1/W3.** ... potential limitations if ... gap between theory and practice?'' ... whether assumptions fully hold ...
>
> A  limitation is that the theory uses abstractions discussed in Q2 to make sequential regulation tractable. If real environments deviate from these conditions, the most likely effect is conservativeness, not catastrophic failure: the proxy $\hat{R}_t^+(c)$ may overestimate true risk, causing earlier intervention, heavier penalties, or slower recovery declaration. In practice, this means CORAL may trade some short-term reward for stronger safety.
>
> However, this gap is mitigated by the design itself. CORAL updates the continuous exposure state $D_t$ online and computes the penalty in Eq.~(12) directly from empirically observed violations. Thus, assumption mismatch may loosen theoretical tightness, but the underlying regulation mechanism remains valid and adaptive.
>
> We also examined this theory–practice gap empirically. On real datasets, CORAL improves EC/PS while preserving utility; in the 100-step closed-loop evaluation, it maintains the strongest long-horizon reward–coverage frontier; and in the amplified echo-chamber setting, it remains stable under stronger feedback loops. We will add this limitation discussion explicitly in the revision.

---

> > ### Author Rebuttal · Reviewer_BwWF · 2026-04-04
> >
> > I appreciate the authors' responses regarding my previous concerns.
> >
> > However, my key concerns remain unresolved. This is reflected in the following two aspects:
> > *   Although the authors have explained the function of each hyperparameter, this does not alter the fact that, in practice, deploying CORAL across different application scenarios requires significant effort to tune the hyperparameters in order to ensure effectiveness. A constructive direction for improvement would be to provide valuable insights and guidance on hyperparameter configuration tailored to various application scenarios (e.g., different recommendation environments).
> >
> > *   Given that a major contribution of this submission lies in improving the recommendation ecosystem, it is essential to deploy the proposed method within a real-world industrial recommendation system and conduct comprehensive, long-term A/B testing. Evaluations on benchmark datasets cannot be directly aligned with them, as these datasets are often no longer consistent with real-world recommendation environments due to extensive preprocessing.
> >
> > Therefore, I stand by my initial assessment.

---

> > > ### Author Response · Authors · 2026-04-07
> > >
> > > We thank the reviewer for their feedback. Below, we address the remaining concerns:
> > >
> > > > ..guidance on hyperparameter configuration tailored to various application scenarios...
> > >
> > > We extend the functionality of hyperparameters in W1 to their  operational roles and practical, scenario-dependent interpretation as follows:
> > >
> > > (i) $\lambda_{\max}$: this threshold should be adjusted based on domain characteristics. For platforms with persisting user interests (e.g., gaming), repeated interactions often reflect stable preference rather than harmful saturation in the Hawkes-intensity sense. In such cases, a slightly larger $\lambda_{\max}$ helps avoid over-triggering regulation on benign repetition.
> > >
> > > (ii) $\delta$: this should be set according to the platform’s tolerance for saturation risk. Since $\delta$ controls the confidence level of the uncertainty-aware risk bound, smaller values yield tighter (more conservative) bounds and therefore stronger regulation near the saturation boundary, while larger values allow more flexibility when occasional violations are acceptable.
> > >
> > > (iii) $\kappa$: this should be set based on the desired exploration--exploitation balance. As $\kappa$ increases, the policy explores more aggressively; as it decreases, the policy becomes more exploitative. In practice, $\kappa$ can be calibrated using the reward--coverage trade-off, where the Pareto-efficient operating point reflects an appropriate balance between long-horizon utility and exposure diversity.
> > >
> > > To make this more concrete, we will add the following suggestive application-oriented configuration table in the revision:
> > >
> > > | **Application scenario** | $\lambda_{\max}$ | $\kappa$ | $\delta$ | **Suggested rationale** |
> > > |---|---:|---:|---:|---|
> > > | Short-video / news feeds | lower | moderate--high | lower--medium | Exposure concentration can escalate quickly; earlier regulation and stronger exploration help avoid rapid confinement into narrow content clusters. |
> > > | E-commerce / conversion-driven browsing | medium | moderate | medium | Repeated within-category interactions may reflect genuine purchase intent rather than harmful saturation, so regulation should not trigger too early. |
> > > | Gaming / stable-interest platforms | higher | low--moderate | medium | Users often sustain repeated interest in a narrow genre over longer periods, so a more permissive saturation threshold avoids over-regulating benign persistence. |
> > > | Sensitive / high-safety-cost domains (e.g., health-related content) | lower | moderate | lower | The goal is to reduce missed risky saturation states; a lower $\delta$ makes the controller more conservative, accepting more intervention in exchange for stronger protection. |
> > >
> > > This table offers guidance on hyperparameter configuration tailored to various application scenarios.
> > >
> > > > ...A/B testing... real world environment change...
> > >
> > > We agree while industrial A/B testing could further validate CORAL, it typically requires access to production systems, live user traffic, and platform-specific logs, which are generally unavailable in an academic setting due to operational and intellectual-property constraints. We therefore view A/B testing and industrial deployment as an important future direction.
> > >
> > > Within the scope of academic validation, we used the strongest feasible alternatives: **real-world datasets, multi-step closed-loop evaluation, robustness analysis under amplified echo-chamber feedback, and theoretical guarantees.** In particular, the closed-loop protocol in Sections 5.2.2 and 5.2.3 directly evaluates long-horizon feedback _(which simulates real world settings)_ dynamics by allowing CORAL’s decisions to influence future states. Across these settings, CORAL consistently maintains the strongest reward-coverage frontier and has flat regulated Hawkes-intensity trajectory.
> > > We believe that, within the scope of academic research, this combination of our theoretical guarantees and this closed-loop simulation provides strong evidence of CORAL's practicality.
> > >
> > > We sincerely hope we addressed the reviewer's concerns and thank them for their constructive feedback, helping us strengthen this work.

---

### Decision · Program_Chairs · 2026-04-30

**Decision:**

Accept (regular)

**Comment:**

CORAL tackles a meaningful and underaddressed problem—feedback-driven exposure concentration in recommender systems—by framing it as a constrained sequential decision problem with uncertainty-aware risk control. The Hawkes-inspired saturation state, UCB-based intervention rule, and finite-sample theoretical guarantees represent a coherent and principled contribution. Two reviewers recommended weak accept, and a third upgraded after the rebuttal addressed scalability, taxonomy sensitivity, and the neutral-category assumption.
The most substantive concern came from Reviewer zp4F, who correctly identified that Theorem 1 bounds historical average risk while the algorithm targets current decision-time risk, and that the Theorem 3 proof sketch introduces additional assumptions not present in the main paper. These are genuine theoretical gaps that the rebuttal addressed with additional derivations but not with a complete, clean proof. Reviewer BwWF's insistence on industrial A/B testing is reasonable but sets a bar beyond what academic submissions can typically meet.
On balance, the problem formulation is novel, the empirical evaluation is broad and includes closed-loop simulation, and the theoretical issues—while real—appear fixable in revision rather than fundamental. The authors should incorporate the corrected theorems, relaxed neutral-category assumption, and hyperparameter guidance in the final version. Borderline accept is recommended.